# A novel strategy to generate immunocytokines with activity-on-demand using small molecule inhibitors

Giulia Rotta [1,2], Ettore Gilardoni[1], Domenico Ravazza[1], Jacqueline Mock[1], Frauke Seehusen[3], Abdullah Elsayed [1,4], Emanuele Puca[1,5], Roberto De Luca[1], Christian Pellegrino [6], Thomas Look[7], Tobias Weiss [7], Markus G Manz [6], Cornelia Halin [4], Dario Neri [1,4,5✉] & Sheila Dakhel Plaza [1✉]

## Abstract

Cytokine-based therapeutics have been shown to mediate objective responses in certain tumor entities but suffer from insufficient selectivity, causing limiting toxicity which prevents dose escalation to therapeutically active regimens. The antibody-based delivery of cytokines significantly increases the therapeutic index of the corresponding payload but still suffers from side effects associated with peak concentrations of the product in blood upon intravenous administration. Here we devise a general strategy (named "Intra-Cork") to mask systemic cytokine activity without impacting anti-cancer efficacy. Our technology features the use of antibody-cytokine fusions, capable of selective localization at the neoplastic site, in combination with pathway-selective inhibitors of the cytokine signaling, which rapidly clear from the body. This strategy, exemplified with a tumor-targeted IL12 in combination with a JAK2 inhibitor, allowed to abrogate cytokine-driven toxicity without affecting therapeutic activity in a preclinical model of cancer. This approach is readily applicable in clinical practice.

**Keywords** Cancer Immunotherapy; Tumor Targeting; Antibody-Cytokine Fusions; Tolerability; Small Molecule Inhibitors
**Subject Categories** Cancer; Immunology; Pharmacology & Drug Discovery

## Introduction

Cytokines are potent modulators of anti-cancer immunity and some recombinant cytokine products have been used for therapeutic applications (Berraondo et al, 2019). For example, recombinant human Interleukin 2 (IL2), Interferon alpha (IFN-α), and tumor necrosis factor (TNF) have gained marketing authorization for the treatment of certain tumor types (Fyfe et al, 1995; Eggermont et al, 1996; Minutilli and Feliciani, 2012), but their clinical use is limited by insufficient activity and substantial

toxicity, even at low (sub-milligram) doses. Interleukin 12 (IL12) has been proposed as one of the most promising cytokines for anti-cancer therapy, but recombinant preparations of this biopharmaceutical could not be used at doses higher than 0.5 µg/kg/day and caused deaths in Phase II clinical trials (Cohen, 1995; Atkins et al, 1997; Car et al, 1999; Haicheur et al, 2000; Lenzi et al, 2007). However, objective responses were observed in several tumor entities (i.e., advanced renal cell carcinoma, metastatic melanoma, T-cell lymphoma, and colon cancer) (Atkins et al, 1997; Bajetta et al, 1998; Portielje et al, 1999; Rook et al, 1999), suggesting that improved IL12-based strategies may be efficacious for fighting aggressive cancer types.

Antibody-cytokine fusions (also called immunocytokines), capable of preferential localization at the tumor site, have been proposed as a novel class of cytokine-based therapeutics with an improved therapeutic index (Reisfeld and Gillies, 1996; Penichet and Morrison, 2001; Epstein et al, 2003; Kontermann, 2012; Neri, 2019). In mice, the targeted delivery of IL2, TNF, and IL12 led to a potentiation of the cytokine payload as a result of increased density and activity of tumor resident T cells and NK cells, capable of neoplastic cell recognition (Halin et al, 2002; De Luca et al, 2017; Puca et al, 2020; Weiss et al, 2020; Look et al, 2023). Immunocytokines based on these payloads have progressed to late-stage clinical trials with promising results in difficult-to-treat cancer indications, including metastatic melanoma (Eigentler et al, 2011; Weide et al, 2014; Danielli et al, 2015), high-risk basal cell carcinoma (NCT03567889), glioblastoma (Look et al, 2023) and acute myeloid leukemia (Berdel et al, 2022). Our group has worked extensively on cytokine fusions based on the fully human L19 antibody, specific to the alternatively spliced EDB domain of fibronectin. This stromal antigen is strongly expressed in the majority of aggressive solid tumors, lymphomas, and leukemias while exhibiting an extremely restricted expression in normal adult tissues (Carnemolla et al, 1989; Castellani et al, 1994; Hooper et al, 2022). The tumor-homing properties of the L19 antibody have been quantified both in mice and in patients using Nuclear Medicine techniques (Erba et al, 2012; Poli et al, 2013). We have consistently used antibody fragment formats to limit the circulation time of the corresponding immunocytokines in blood, with the aim of reducing

[1]Philochem AG, CH-8112 Otelfingen, Switzerland. [2]Department of Cellular, Computational, and Integrative Biology (CIBIO), University of Trento, 38123 Trento, Italy. [3]Laboratory for Animal Model Pathology (LAMP), Institute of Veterinary Pathology, University of Zurich, CH-8057 Zurich, Switzerland. [4]Institute of Pharmaceutical Sciences, ETH Zurich, CH-8093 Zurich, Switzerland. [5]Philogen S.p.A, 53100 Siena, Italy. [6]Department of Medical Oncology and Hematology, University Hospital Zurich and University of Zurich, CH-8091 Zurich, Switzerland. [7]Department of Neurology, Clinical Neuroscience Center, University Hospital Zurich and University of Zurich, CH-8091 Zurich, Switzerland. ✉E-mail: dario.neri@pharma.ethz.ch; sheila.dakhel@philochem.ch

cytokine-driven adverse events. However, clinical experience with L19-IL2, L19-TNF, and L19-IL12 has shown that these products typically cause transient toxicity (e.g., flu-like symptoms, hypotension, nausea) at early time points after intravenous administration, in concomitance with high systemic cytokine levels (Johannsen et al, 2010; Spitaleri et al, 2013; Danielli et al, 2015; Weiss et al, 2020; Look et al, 2023). The adverse events typically disappear when the concentration of the payload falls below a suitably low threshold. It would thus be desirable to devise a therapeutic strategy that retains the activity of tumor-homing cytokine products while sparing the patient from unacceptable toxicity, associated with the peak concentration of the product in blood at early time points.

In this work, we present a novel approach, called "Intra-Cork" technology, which relies on the use of pathway-selective small molecule inhibitors capable of transiently inhibiting the intracellular cytokine signaling in the periphery. The use of specific cognate inhibitors preserves the long residence time and anti-tumor activity of the cytokine moiety within the tumor mass, as small molecules rapidly clear from circulation. Here, the "Intra-Cork" strategy has been exemplified with L19-IL12 in combination with Ruxolitinib, a commercially available JAK2 inhibitor. L19-IL12 (fully human fusion protein) is currently being investigated in a Phase I clinical trial for the treatment of advanced solid carcinomas (NCT04471987). Ruxolitinib (INCB018424) is approved for the treatment of myelofibrosis, polycythemia vera, and steroid-refractory acute graft-versus-host-disease (fda & cder, 2019). All these disorders are characterized by excessive inflammation driven by a massive activation of immune cells and cytokine production (Ferrara, 1993; Tefferi et al, 2011; Pourcelot et al, 2014) and therefore, the use of Ruxolitinib has been considered for the management of hyperinflammatory conditions.

Our data show that the use of Ruxolitinib enabled the administration of high doses of the L19-mIL12 (carrying the murine payload) that would have otherwise not been tolerated in mice. Importantly, the transient inhibition of JAK2 helped to control massive cytokine release, preserve liver integrity, and did not interfere with the potent anti-tumor activity of L19-mIL12 and its immunological effects at the neoplastic site. We have previously shown that the specific RIPK1 inhibitor (GSK'963) of tumor necrosis factor signaling substantially increased the therapeutic index of L19-TNF, while ibuprofen did not reduce treatment-related toxicity (Dakhel et al, 2019). Overall, this study shows that the "Intra-Cork" strategy may represent a broadly applicable approach to overcome dose-limiting immune-related adverse events associated with high doses of tumor-homing cytokine fusions.

## Results

### In vitro screening of small-molecule kinase inhibitors identifies Ruxolitinib as a potent modulator of L19-IL12 signaling

We devised a strategy, so-called Intra-Cork, to retain the anti-cancer activity of targeted IL12 while sparing the body from unnecessary toxicity. The "Intra-Cork" term was born with the analogy of a glass bottle, representing the activity of an immunocytokine (e.g., L19-IL12), whose content (the activity of the payload) is contained by a cork (e.g., Ruxolitinib, the small molecule inhibitor). Given the fact that this particular cork acts on the cytokine receptor intracellular signaling pathway, the term "Cork" was paired with the adjective "Intra": Intra-Cork. The approach crucially relies on (i) the preferential and long tumor accumulation of an antibody-IL12 fusion protein and (ii) the short half-life of a small molecule inhibitor. Shortly after intravenous administration, L19-IL12 has the potential to interact with its cognate receptor on lymphocytes in the body before accumulating at the tumor site. JAK2 inhibitors can transiently mask the intracellular signaling of the cytokine and the resulting toxicity, de facto creating a rapid and reversible pharmacological ON/OFF switch that localizes the activity of the payload to the tumor site while sparing healthy tissues (Fig. 1A). L19-mIL12 (murine IL12) and L19-IL12 (clinical stage product) were produced in a stable and monomeric form as assessed by SDS-PAGE and size exclusion chromatography (Figs. 1B,C and EV1A,B). To identify molecules able to block L19-IL12 signaling, we screened a panel of JAK inhibitors (JAKi) on human NK-92 cells, and IFN-$\gamma$ release was quantified as a readout of IL12 activity. As a result, Ruxolitinib, acting as a potent JAK2i, could strongly block IFN-$\gamma$ production upon L19-IL12 stimulation. Ruxolitinib exhibited an IC$_{50}$ of 42 nM, showing remarkable superiority in potency compared to inhibitors with similar selectivity, such as Baricitinib (IC$_{50}$ = 462 nM) or Tofacitinib (IC$_{50}$ = 3900 nM) (Fig. 1D). By contrast, neither a JAK3-specific inhibitor (Ritlecitinib) nor a commonly used immune-modulatory drug (Dexamethasone) could block IL12 activity in vitro, emphasizing the importance of using pathway-selective inhibitors. Based on these results, our work subsequently focused on Ruxolitinib as "Intra-Cork" drug to mitigate L19-IL12 treatment-associated toxicities.

### Ex vivo MS-based biodistribution of Ruxolitinib shows a rapid clearance of the drug

Ruxolitinib is an orally bioavailable inhibitor showing a linear profile in humans over a dose range of 5 to 200 mg (Shi et al, 2011). The drug exhibits near-complete oral absorption achieving maximal plasma concentration (Cmax) at 1–2 h post-administration and a terminal elimination half-life of approximately 3 h (5.8 h for Ruxolitinib + metabolites) (Shilling et al, 2010). In this work, Ruxolitinib was administered by subcutaneous injections in MC-38 tumor-bearing mice. In an ex vivo biodistribution study, the small molecule was rapidly absorbed into the blood, tumor, and healthy tissues (Tmax ≤10 min) and quickly metabolized or excreted from the body, showing a half-life of ≈ 1 h. Importantly for our project, Ruxolitinib did not selectively accumulate in any of the analyzed organs (Figs. 1E and EV1C) and had a blood clearance profile that matched the one described in humans (Shilling et al, 2010). Based on these results, pre-administration of Ruxolitinib 10 min prior to the immunocytokine was adopted for subsequent in vivo efficacy studies.

### Pre-treatment with Ruxolitinib enhances the tolerability of L19-mIL12

We then investigated whether the use of Ruxolitinib, acting as a signaling modulator of L19-IL12, enhanced the tolerability of the immunocytokine while preserving its anti-cancer efficacy. For this

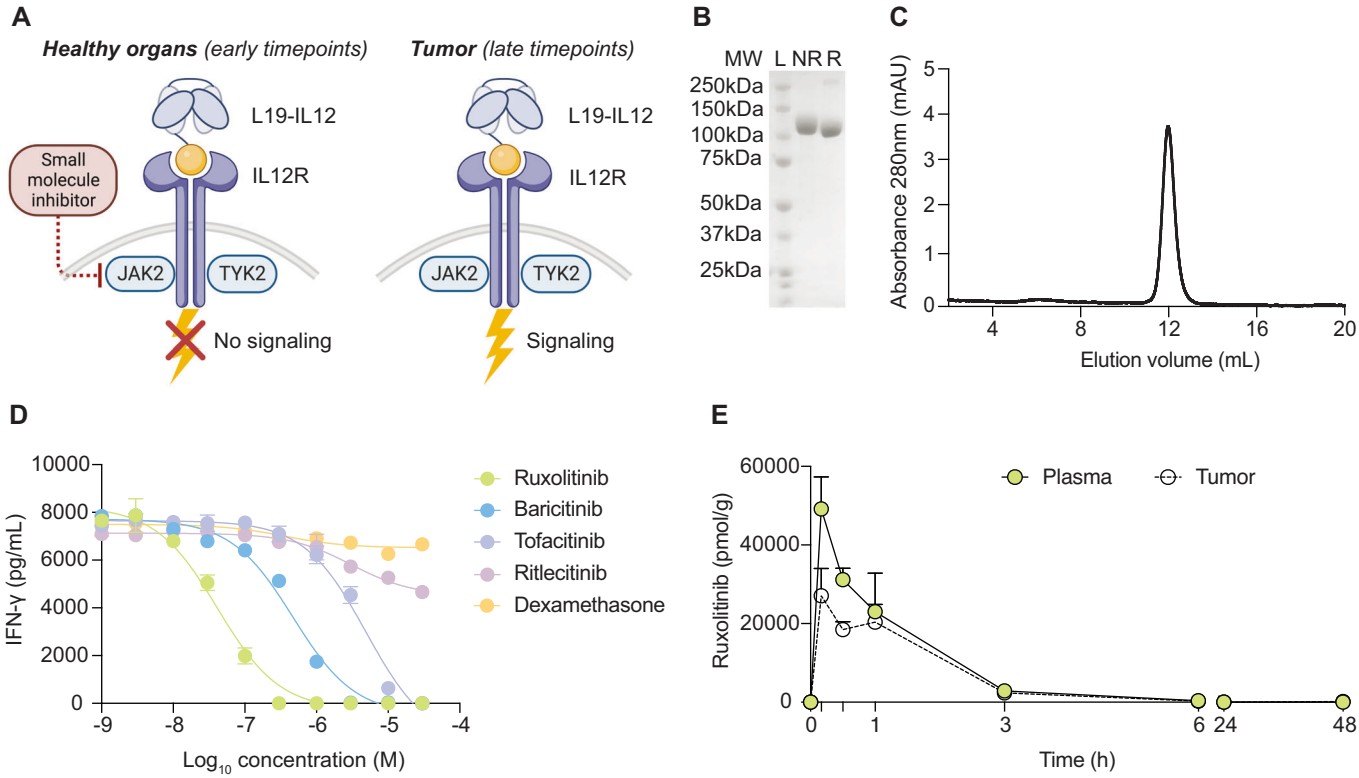

**Figure 1. Ruxolitinib is selected as a potent modulator of L19-mIL12 activity.**

(A) Schematic representation of the Intra-Cork technology approach. The pre-administration of a kinetically-matched inhibitor can transiently block L19-IL12 activity in blood shortly after administration. Anti-tumor activity is not impaired by virtue of the long residence time of the targeted cytokine at the neoplastic site and the fast clearance of the inhibitor from circulation. (B,C) Quality control analyses of L19-mIL12 fusion protein assessed by SDS-PAGE (B) and size exclusion chromatography (C). MW = molecular weight; L = ladder; NR = non-reducing conditions; R = reducing conditions. (D) Human NK-92 cells were stimulated in vitro with 10 ng/mL of L19-IL12 for 24 h in the presence of increasing concentrations of several small molecule drugs. IFN-γ levels in the supernatant were determined by ELISA. Data represent mean ± SD of triplicate wells for each condition. (E) Quantitative ex vivo MS-based biodistribution of Ruxolitinib at different time points after a single dose in MC-38 tumor-bearing mice (75 mg/kg, s.c). Results are expressed as picomoles per gram of tissue (mean ± SD, $n = 3$ mice per time point). Source data are available online for this figure.

purpose, an L19-mIL12 dose escalation study in combination with Ruxolitinib was carried out in C57BL/6 mice bearing MC-38 tumors. C57BL/6 is a strain that has been described to be particularly susceptible to IL12 treatment, as it displays a Th-1 skewed immune response (Nakamura et al, 2000). In our studies, Ruxolitinib was pre-administered subcutaneously at a fixed dose of 75 mg/kg, which had previously been shown to be safe in mice (Quintás-Cardama et al, 2010). L19-mIL12 effectively controlled tumor growth at all doses, and pre-treatment with Ruxolitinib did not interfere with the anti-tumor activity (Fig. 2A–C). To assess treatment-related toxicity, we monitored the body weight loss (BWL) daily and consistently at the same time to exclude physiological body weight fluctuations (Fig. 2D–F). Mice injected with L19-mIL12 lost weight in a dose-dependent fashion reaching the peak of BWL around two days after the last injection. At the highest dose (1.2 mg/kg), 40% of the mice showed signs of severe toxicity, meeting the endpoint for animal termination (BWL > 15%). By contrast, the combination of L19-mIL12 with Ruxolitinib allowed complete protection from BWL at the recommended dose (0.6 mg/kg) and partial protection at increasing doses. Based on these encouraging results, we then sought to investigate whether the pre-treatment with Ruxolitinib could further escalate the dose of

L19-mIL12 beyond its maximal tolerated dose (MTD). Administration of very high doses of L19-mIL12 (2.4 mg/kg) in combination with Ruxolitinib further potentiated anti-tumor activity, achieving cures in a proportion of mice (1/5 CR) (Fig. 2G). Moreover, treatment with L19-mIL12 alone led to severe toxicity, with 100% of the animals having to be euthanized already after the second injection. By contrast, pre-treatment with Ruxolitinib significantly increased the overall survival of the mice treated with otherwise lethal doses of L19-mIL12 (Fig. 2H,I). Altogether, these data indicate that the transient signaling inhibition using Ruxolitinib can represent an avenue to improve the therapeutic window of L19-mIL12.

## Pre-treatment with Ruxolitinib does not impact the tumor remodeling induced by L19-mIL12

To investigate whether Ruxolitinib pre-treatment interferes with the therapeutic activity of L19-mIL12, we characterized immunological responses in MC-38 tumor-bearing mice at the end of treatment. L19-mIL12 treatment, alone or in combination with Ruxolitinib, equally increased the density of CD4[+] and CD8[+] T cells within the tumor compared to the saline group (Fig. 3A). The

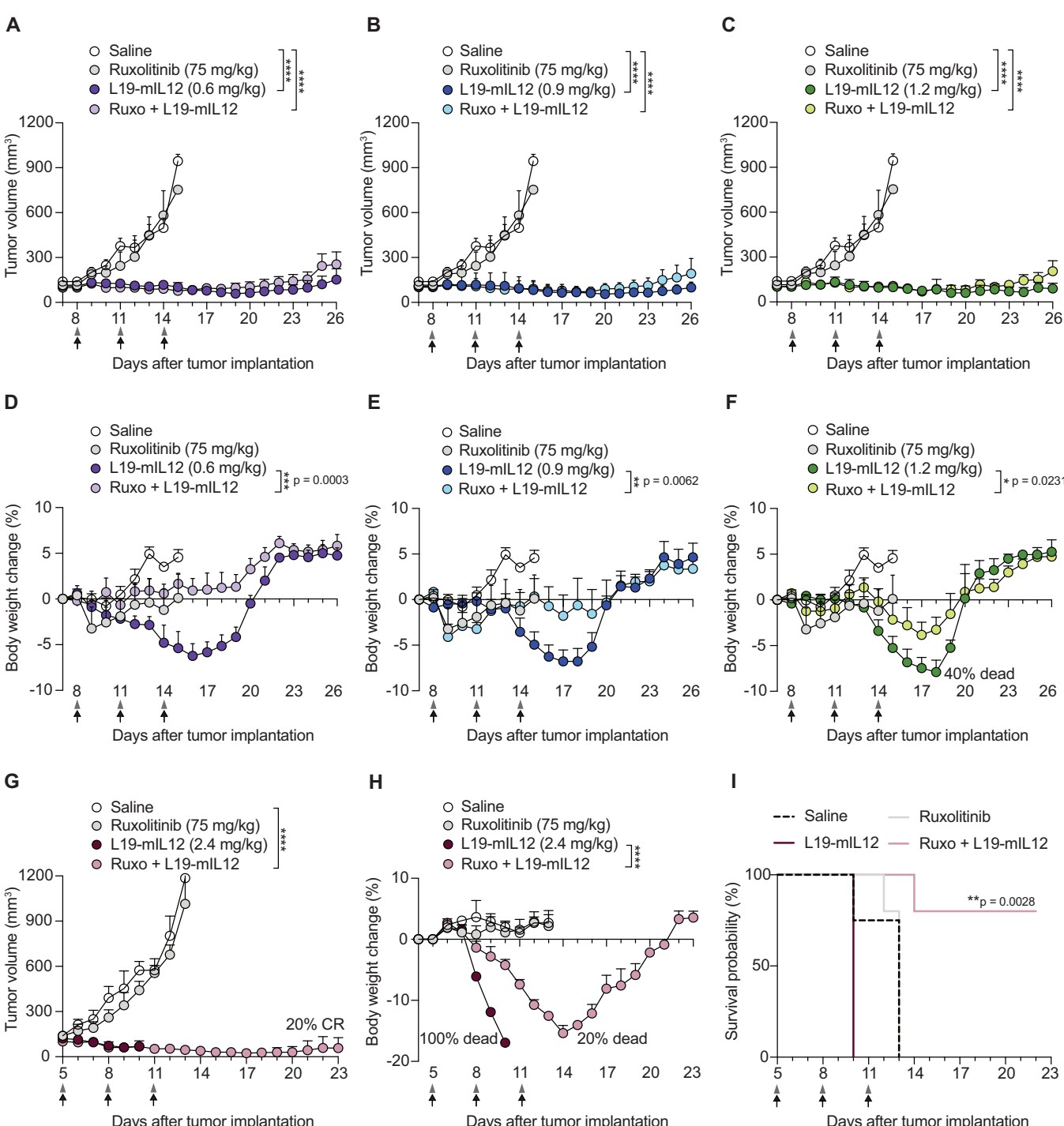

**Figure 2. Pre-treatment with Ruxolitinib enhances the tolerability of L19-mIL12.**

(A–C) Tumor growth curves of C57BL/6 mice inoculated with $1 \times 10^6$ MC-38 cells. Treatment started when the tumors reached 100 mm³ on average ($n = 5$ mice per group). Mice received different doses of L19-mIL12 alone (i.v, black arrows) or in combination with a pre-treatment of Ruxolitinib (75 mg/kg, s.c, gray arrowheads). (D–F) Treatment tolerability was assessed by daily body weight changes of the mice ($n = 5$ mice per group). (G–I) Tumor growth (G), body weight change (H), and Kaplan–Meier survival curve (I) of MC-38 tumor-bearing mice receiving ultra-high doses of the L19-mIL12 (2.4 mg/kg intravenous administration) with or without a pre-treatment with Ruxolitinib. In all in vivo studies, a full therapy cycle was given, including three injection days, every 72 h (black arrows). CR, complete response ($n = 5$ mice per group). Data information: in (A–H) data represent mean tumor volume and body weight change (%) ± SEM. Two-way ANOVA analysis (*$p < 0.05$; **$p < 0.01$; ***$p < 0.001$; ****$p < 0.0001$). Source data are available online for this figure.

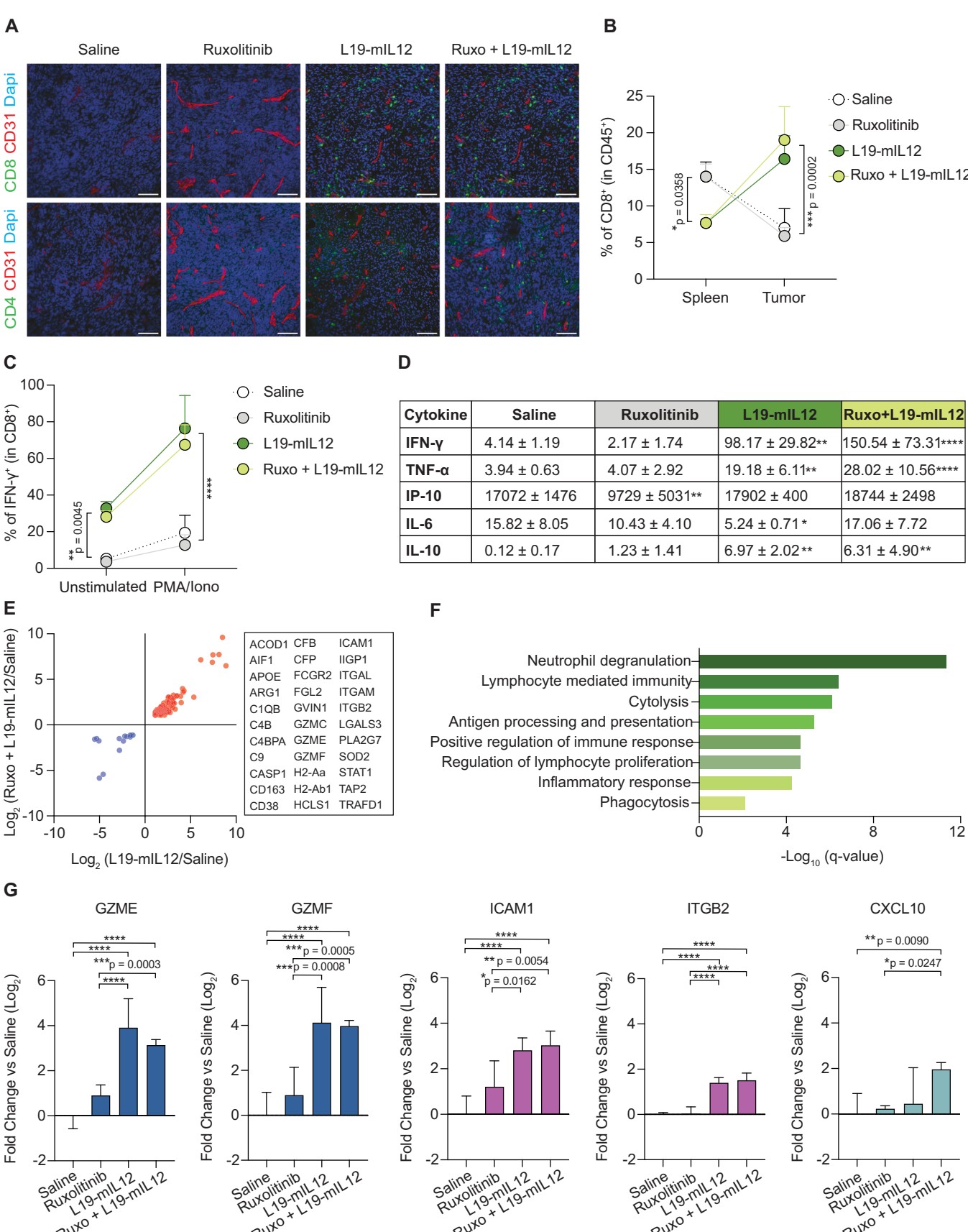

**Figure 3.** **Pre-treatment with Ruxolitinib does not impact the tumor remodeling induced by L19-mIL12.**

MC-38 tumor-bearing mice were euthanized 24 h after the third injection of either saline, Ruxolitinib (75 mg/kg, s.c), L19-mIL12 (1.2 mg/kg, i.v), or Ruxolitinib pre-administered before the L19-mIL12. (**A**) Ex vivo immunofluorescence analysis of tumor immune infiltrates was evaluated using markers specific for CD4$^+$ T cells (CD4) and CD8$^+$ T cells (CD8) (green). Blood vessels were stained with an anti-CD31 antibody (red) and nuclei with DAPI (blue). The images are a representative area of the entire sample. Magnification: 10×; scale bars = 100 μm. (**B,C**) Flow cytometry analyses were performed in the same mice. Percentage of CD8$^+$ T cells among CD45$^+$ cells in tumors and spleens (**B**) and percentage of IFN-γ$^+$ cells among CD8$^+$ T cells in tumors (**C**) with or without PMA/Iono stimulation (20 ng/mL PMA and 1 μg/mL Ionomycin for 4 h). (**D**) Quantification of cytokine levels in the tumor samples using a Multiplex immunoassay (concentration expressed in pg/mL). (**E–G**) Proteomic analysis was carried out in the tumor samples. Scatterplot comparing changes in protein expression after L19-mIL12 treatment, alone or in combination with a Ruxolitinib pre-treatment relative to the saline group (**E**). Up to 100 proteins were selected based on FDR < 0.01 and magnitude of change >2-fold. Red and blue dots represent commonly up- and down-regulated proteins. The names of some proteins involved in immune-related pathways are listed. Pathway enrichment analysis was used to identify significantly impacted biological pathways after IL12 treatments (**F**). Expression of granzymes (blue), integrins/integrin ligands (violet), and chemokines/cytokines (light blue) represented as fold change compared to the saline group (**G**). Data information: in (**B–G**), data represent mean or fold-change ± SD ($n = 3$–5 mice per group). One-way ANOVA analysis (*$p < 0.05$; **$p < 0.01$; ***$p < 0.001$; ****$p < 0.0001$). Source data are available online for this figure.

higher proportion of the T cell subset in the tumor immune infiltrates was accompanied by a reduction in secondary lymphoid organs, indicating the establishment of a tumor-redirected immune response upon L19-mIL12 treatment (Figs. 3B and EV2A,B). Phenotypic characterization of T cells in the spleen and blood revealed a skewing towards a central-memory and effector-memory phenotype compared to the saline group, with no significant differences between L19-mIL12 treatments (Fig. EV2C,D). IL12 is well-known to induce anti-tumor responses in association with IFN-γ production (Smyth et al, 2000). Importantly, Ruxolitinib pre-treatment did not impact the cytotoxic capacity of tumor-infiltrating lymphocytes, as evidenced by high levels of intracellular IFN-γ (Fig. 3C) and pro-inflammatory cytokines (Fig. 3D). To complement these findings, a proteomic analysis was performed on the treated tumors allowing us to identify more than 5700 proteins. Differentially expressed proteins between the control groups and L19-mIL12 groups were observed (Fig. EV3A). Ruxolitinib alone had little to no impact on the tumor proteome compared to saline. On the contrary, L19-mIL12 treatments markedly affected the proteome in the tumors resulting in about 100 proteins being up- or down-regulated (Fig. 3E), all associated with inflammatory processes (Fig. 3F). Among the significantly up-regulated proteins, several granzymes (GZMC, GZME, GZMF), integrins/integrin ligands (ICAM1, ITGB2, ITGAM, ITGAL), chemokines/cytokines (CXCL10, CCL8, IL16) and immune-cell markers (CSF1R, CD38, CD14, CD163, LCN2) could be identified (Figs. 3G and EV3B), indicating that L19-mIL12 was able to turn the tumors into immunologically active sites also when used in combination with Ruxolitinib. These findings may be of clinical relevance, as they suggest that short-term treatment with Ruxolitinib would not impair the anti-tumor effect of immunotherapeutic drugs.

## Pre-treatment with Ruxolitinib minimizes the "on-target off-tumor" activity of L19-mIL12

Shortly after IL12 infusion, patients may experience high-grade toxicities, including fever and flu-like symptoms, which most frequently occur after the first therapy administration (Robertson et al, 1999; Greiner et al, 2021). Such acute side effects may potentially be associated with a "cytokine storm," consisting of a rapid and transient elevation of pro-inflammatory cytokines in the blood due to systemic lymphocyte activation. We sought to understand how the transient inhibition of the IL12 signaling cascade could mitigate cytokine release. For this purpose,

circulatory IFN-γ levels were monitored over time after a single injection of L19-mIL12 alone or in combination with Ruxolitinib. L19-mIL12 elicited an acute IFN-γ response shortly after injection, followed by a second wave of IFN-γ release at around 24 h (Fig. 4A). Importantly, Ruxolitinib pre-treatment could completely abolish IFN-γ production up to 6 h after L19-mIL12 treatment, while it could not lower cytokine release at later time points, in line with its rapid clearance. To further prove the signaling inhibition exerted by Ruxolitinib upon L19-mIL12 treatment, STAT4 phosphorylation (p-STAT4), a downstream event of IL12/IL12R signaling (Hu et al, 2021), was monitored over time in vivo. We observed a higher frequency of p-STAT4$^+$ cells in the blood of mice receiving the L19-mIL12 alone compared to the combination group. This effect was more evident at early time points, whereas similar levels were detected at later time points (Fig. 4B). Additionally, a panel of cytokines was analyzed in mice receiving a full therapy cycle. Increasing levels of circulating pro-inflammatory cytokines were quantified upon L19-mIL12 treatment, irrespectively of Ruxolitinib pre-administration (Fig. 4C). Altogether, these data suggest that Ruxolitinib can transiently inhibit the signaling induced by L19-mIL12 in vivo and therefore control acute cytokine release. Dose-limiting toxicities associated with the systemic administration of IL12-based products include a decrease in Red Blood Cells (RBC) and peripheral leukocytes, together with marked splenomegaly (Car et al, 1995). It has been postulated that these effects are due to the redistribution of leukocytes into tissues and the enhancement of extramedullary hematopoiesis driven by IL12 (Tare et al, 1995). We, therefore, investigated whether pre-treatment with Ruxolitinib could reduce the "on-target off-tumor" toxicity associated with very high doses of L19-mIL12. Hematological changes and spleen size were analyzed in the treatment groups 24 h after a full therapy cycle. Our data revealed a statistically significant reduction in the absolute RBC count in mice treated with high doses of L19-mIL12 alone and a mild recovery in the combination group (Fig. 4D). Notably, our experimental settings did not fully resemble the leukopenia or neutropenia observed in clinical trials. With regard to secondary lymphoid organs, the splenomegaly induced by L19-mIL12 treatment was significantly reduced by Ruxolitinib pre-treatment despite comparable levels of extramedullary hematopoiesis observed in the pathological analysis of the organ (Fig. 4E,F). It has been postulated that erythrocytes can be trapped in enlarged spleens (Tare et al, 1995) and our findings suggest that Ruxolitinib can be used to modulate the peripheral activity of L19-mIL12.

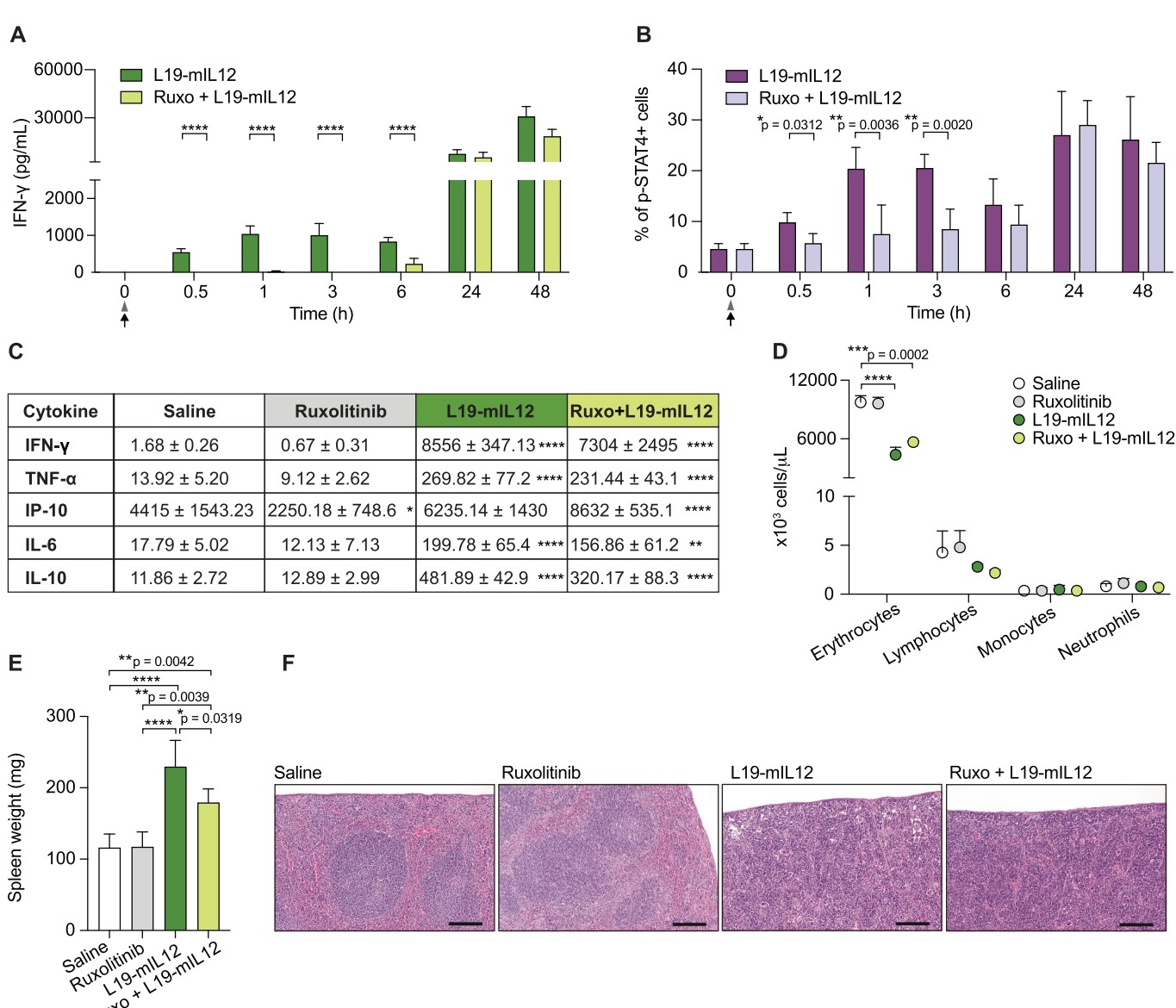

**Figure 4. Pre-treatment with Ruxolitinib minimizes the "on-target off-tumor" activity of L19-mIL12.**

(A,B) MC-38 tumor-bearing mice were euthanized at different time points after a single injection of L19-mIL12 (1.2 mg/kg, i.v, black arrow) alone or in combination with a pre-treatment of Ruxolitinib (75 mg/kg, s.c, gray arrowhead) ($n = 3$ mice per group). Plasmatic levels of IFN-γ quantified over time by ELISA (A). Percentage of phospho-STAT4+ cells among CD45+ cells in the blood at different time points, analyzed by flow cytometry (B). (C–F) MC-38 tumor-bearing mice were euthanized 24 h after the third injection of either saline, Ruxolitinib (75 mg/kg, s.c), or L19-mIL12 (1.2 mg/kg, i.v) alone or in combination with a pre-treatment of Ruxolitinib ($n = 3$ mice per group). Blood samples were processed, and quantification of cytokine levels by Multiplex (C) (concentration expressed in pg/mL) and blood cell counts (D) (expressed in absolute number) were assessed ($n = 3$ mice per group). Spleens from each treatment were weighted (E) ($n = 5$ mice per group). H&E staining in sections of spleens (F). The picture chosen is a representative area of the entire organ. Magnification: 10×; scale bars = 250 µm. Data information: in (A–E), data represent mean ± SD. One-way ANOVA analysis (*$p < 0.05$; **$p < 0.01$; ***$p < 0.001$; ****$p < 0.0001$). Source data are available online for this figure.

## Pre-treatment with Ruxolitinib reduces the hepatotoxicity associated with high doses of L19-mIL12

The liver is one of the main organs associated with dose-limiting toxicities (DLTs) in immunotherapy-treated patients (Zhang et al, 2019; Swanson et al, 2022). Hepatitis and drug-induced liver injury models showed that hepatic damage is frequently associated with immune cell activation and TNF-α and IFN-γ function as the critical cytokines in the pathogenesis (Siwicki et al, 2021). We

investigated the protective effect of Ruxolitinib on the hepatic damage associated with very high doses of L19-mIL12 in mice receiving a full therapy cycle. In fact, the pathological analysis of livers revealed multifocal areas of moderate necrosis as well as moderate lobular and portal infiltrates in mice treated with L19-mIL12. By contrast, degenerative liver changes were surprisingly reduced in the livers of mice receiving the Ruxolitinib pre-treatment, with only mild necrotic alterations (Fig. 5A). In addition, a cumulative histological scoring of the liver (Ishak

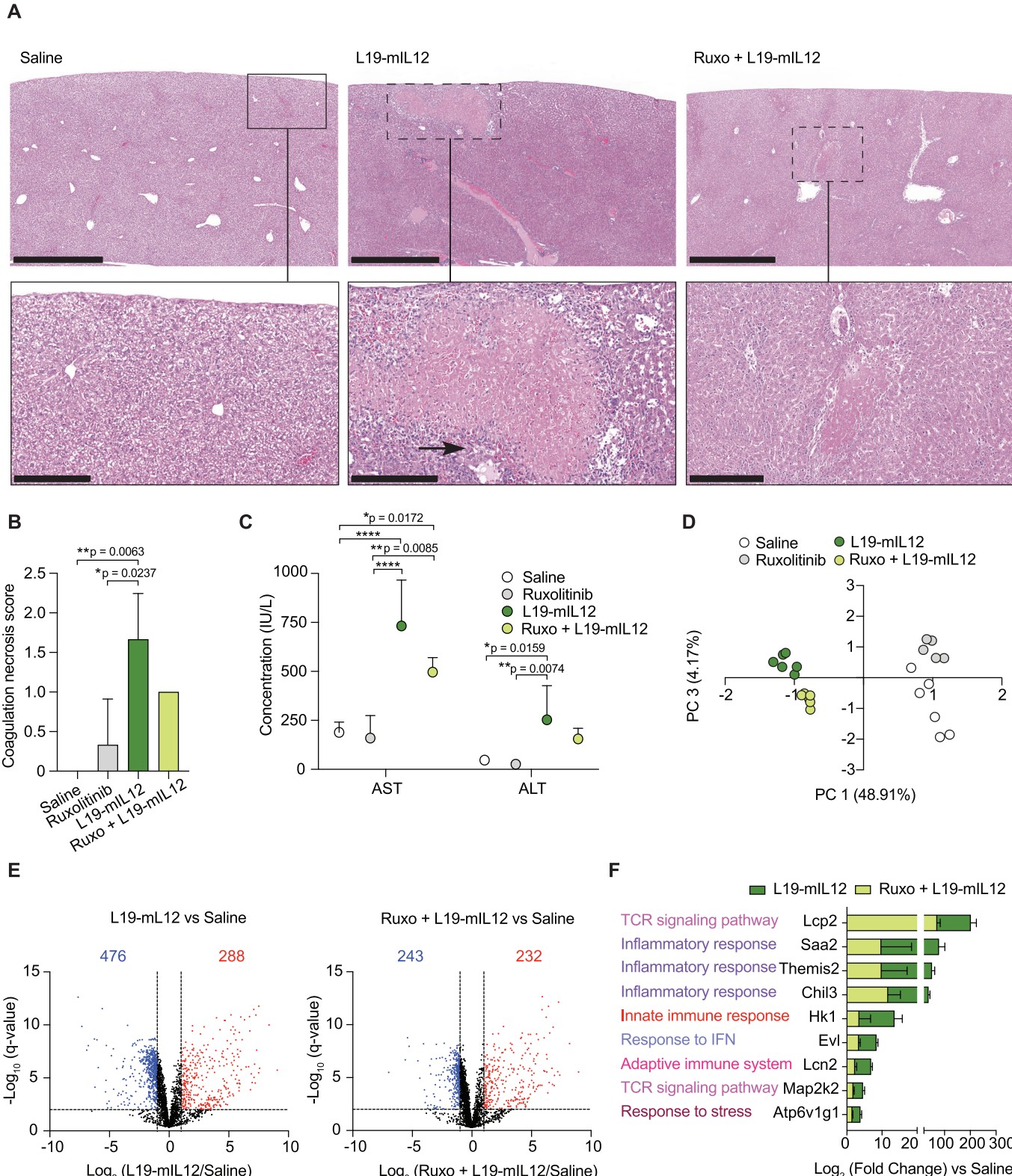

et al, 1995) revealed significant coagulation necrosis in the monotherapy treatment, that was less pronounced in the combination group (Figs. 5B and EV4A). In agreement with these findings, ALT and AST levels in plasma were found to be lower in the combination group (Fig. 5C), accompanied by an increase of

enzymes in the hepatic tissue (Fig. EV4B), indicative of diminished cell damage with membrane disruption. A liver proteomic analysis was carried out to better characterize the differences between the experimental groups. Principal Component Analysis (PCA) allowed clustering of the different treatments based on principal

**Figure 5. Pre-treatment with Ruxolitinib reduces the hepatotoxicity associated with high doses of L19-mIL12.**

MC-38 tumor-bearing mice were euthanized 24 h after the third injection of either saline, Ruxolitinib (75 mg/kg, s.c.), or L19-mIL12 (1.2 mg/kg, i.v) alone or in combination with a pre-treatment of Ruxolitinib ($n = 3$ mice per group). (A–C) H&E staining of liver sections (A) at 10× magnification (scale bars = 1 mm; upper panels) and 20× magnification (scale bars = 250 μm; lower panels). Necrotic lesions are limited by the dotted square, and multifocal periportal inflammatory infiltrates are pointed out with the black arrow. Quantification of coagulation necrosis in the liver (B). Plasmatic concentration of aspartate aminotransferase (AST) and alanine aminotransferase (ALT) liver enzymes (C) ($n = 5$ mice per group). Data represent mean ± SD. One-way ANOVA analysis ($*p < 0.05$; $**p < 0.01$; $****p < 0.0001$). (D–F) Proteomic analysis of livers for the different treatment groups. Principal component analysis (PCA) of livers (D) and volcano plot representation (E) of the proteomic changes in L19-mIL12 monotherapy or in combination with Ruxolitinib, compared to the saline group. Red and blue dots represent significantly up- and down-regulated proteins with FDR < 0.01 and magnitude of change >2-fold. Expression changes in the livers of mice treated with very high doses of the L19-mIL12, with (light green) or without the Ruxolitinib (dark green), were analyzed (F). Data represent fold-change ± SD. Source data are available online for this figure.

component 1 (PC1, explaining roughly 50% of variance) and principal component 3 (PC3, explaining roughly 5% of variance), suggesting that the proteomic landscapes were at least partially distinct (Fig. 5D). The proteomic analysis allowed us to identify more than 3500 proteins in the liver samples. Globally, 288 and 476 proteins were statistically up- and down-regulated, respectively, upon L19-mIL12 treatment. Pre-administration of Ruxolitinib resulted in a decrease of the statistically up- and down-regulated proteins to 232 and 243, respectively (Figs. 5E and EV4C). Biologically relevant pathways involving IL12 activity were identified, including immune system activation and inflammatory response among others, and the expression level of several proteins was found to be significantly reverted upon combination treatment (Fig. 5F). Overall, our data indicate that a single administration of Ruxolitinib can partially protect from hepatotoxicity induced by systemic administration of high doses of L19-mIL12.

### Schedule optimization of Ruxolitinib treatment can completely mask "on-target off-tumor" activity of L19-mIL12

In our preclinical model, we observed that the half-life of the L19-mIL12 after intravenous administration was around 6 h (Fig. 6A). Based on this result and considering the previously described half-life of Ruxolitinib (Fig. 1E), we reasoned that prolonged masking of the L19-mIL12 activity could further improve the safety profile of the immunocytokine. For this, we repeated a tolerability study using high doses of L19-mIL12 in combination with two suitably spaced injections of Ruxolitinib (10 min before and 6 h after the immunocytokine administration). Importantly, a second injection of the IL12 signaling inhibitor allowed improved protection from BWL without impacting the anti-tumor activity of the immunocytokine (Fig. 6B,C). The observed tumor growth retardation observed in this study indicates that the tumor-targeting properties of L19-mIL12 are not affected by the treatment with the Intra-Cork inhibitor. Moreover, the prolonged masking activity of Ruxolitinib completely abolished IFN-γ production in blood for a longer period of time after L19-IL12 treatment and significantly lowered cytokine levels at later time points (Fig. 6D). In peripheral organs, the double injection of Ruxolitinib further reduced splenomegaly (Fig. 6E) and, remarkably, fully protected the liver from coagulation necrosis (Figs. 6F,G and EV5A) induced by high doses of L19-mIL12 treatment. Immunohistochemical evaluation revealed moderate lymphohistiocytic lobular and portal infiltrates in mice treated with L19-mIL12 with a similar number and distribution in the combination group (Fig. EV5B,C). Altogether, these data suggest that repeated dosing, suitably spaced, of the JAK2 inhibitor might

represent an avenue to control immune-related adverse events associated with high systemic levels of L19-IL12, providing a rationale for using the conventional b.i.d. (*bis in die*) schedule of Ruxolitinib.

## Discussion

Cytokine-based cancer therapeutics typically suffer from substantial toxicity shortly after intravenous infusion and insufficient activity, unless the product selectively localizes at the neoplastic site, leading to an activation of tumor resident lymphocytes. The concomitant administration of tumor-homing antibody-cytokine fusions and short-lived selective cytokine signaling inhibitors promises to minimize treatment toxicity without affecting therapeutic potency. This interventional strategy was experimentally demonstrated in an immunocompetent mouse model of cancer using L19-mIL12 and Ruxolitinib, a commercially available JAK2 inhibitor. The corresponding fully human IL12 fusion (L19-IL12) is currently being evaluated in a phase I clinical trial in cancer patients (NCT04471987).

An ideal cytokine-based pharmaceutical would deliver a suitable immunostimulatory payload to the site of disease, helping spare normal tissues. This goal can be facilitated by using antibody-cytokine fusions directed against spliced variants of extracellular matrix components (Schrama et al, 2006; Villa et al, 2008; Young et al, 2014; Müller, 2015; Neri and Sondel, 2016; Hutmacher and Neri, 2019). Among dozens of cytokine payloads, IL2, IL12, and TNF have emerged as some of the most efficient immunomodulatory proteins in preclinical models of cancer (Halin et al, 2002; De Luca et al, 2017; Puca et al, 2020; Weiss et al, 2020; Look et al, 2023). Quantitative biodistribution studies have shown that those radiolabeled immunocytokines directed against the EDB domain of fibronectin preferentially localize to the tumor without being trapped in blood or secondary lymphoid organs (Erba et al, 2012; Poli et al, 2013). Encouraging clinical evidence of exceptional therapeutic activity is emerging in certain indications such as metastatic melanoma (Eigentler et al, 2011; Weide et al, 2014; Danielli et al, 2015), high-risk basal cell carcinoma (NCT03567889), glioblastoma (Look et al, 2023) and acute myeloid leukemia (Berdel et al, 2022). The use of antibody fragments as delivery vehicles is particularly attractive, as they may facilitate a preferential uptake in the neoplastic lesion while being rapidly cleared from the circulation (Carnemolla et al, 2002; Halin et al, 2002; Borsi et al, 2003; De Luca et al, 2017; Ongaro et al, 2019). Cytokine therapeutics devoid of specific cancer recognition modules (e.g., Fc-fusions, PEGylated or recombinant cytokines)

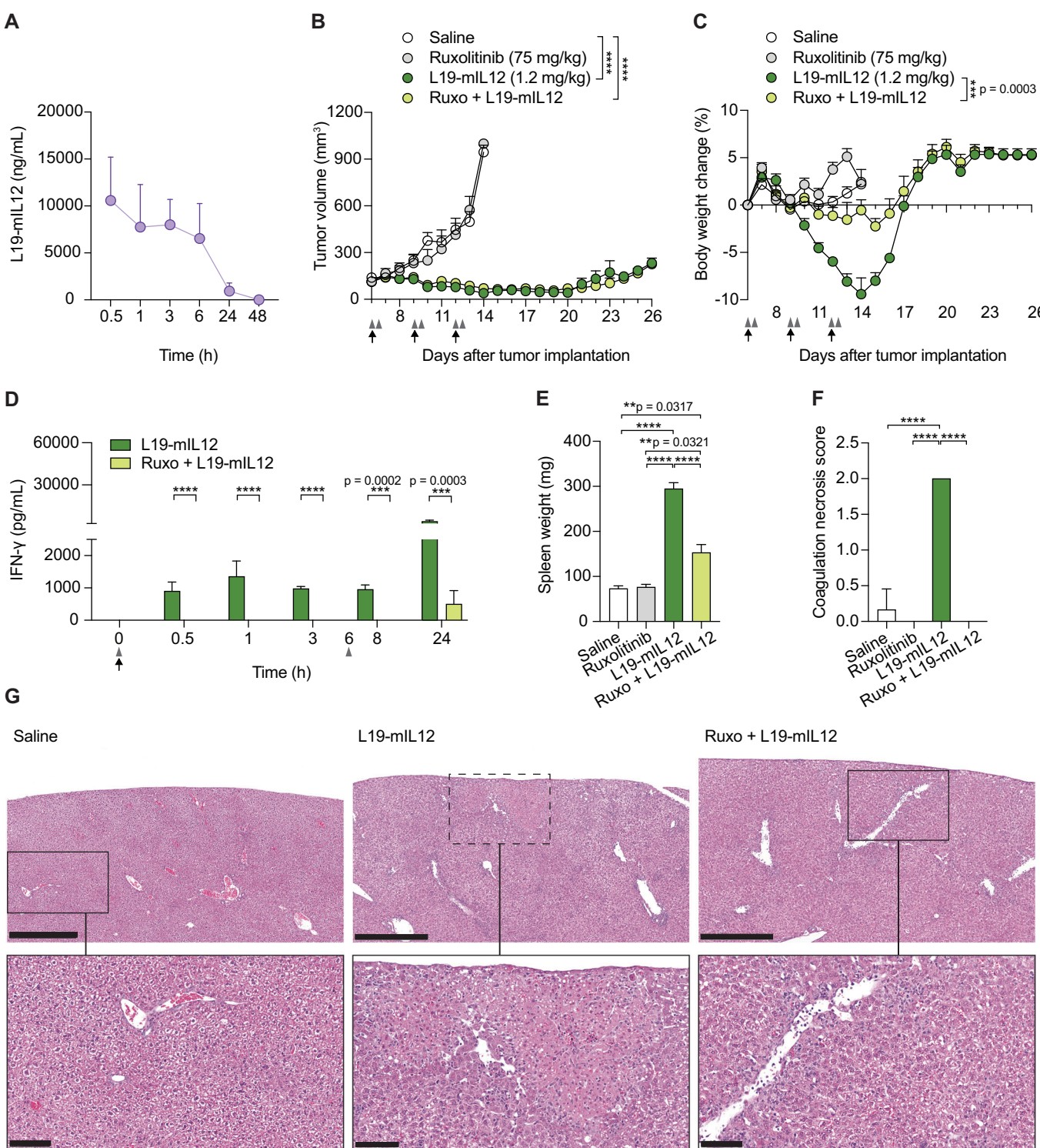

typically do not preferentially home to the tumor and exhibit inferior therapeutic activity in comparative preclinical studies (Halin et al, 2002).

Efforts to achieve "activity-on-demand" have included the use of masking peptides with proteolytic cleavage site (Hsu et al, 2021; Mansurov et al, 2022; Nirschl et al, 2022; Xue et al, 2023), assembly-based reconstitution of cytokine activity at the tumor site (Venetz et al, 2016; Mock et al, 2020), and drug-inducible OFF-

switch of the cytokine (Marchand et al, 2023). The "Intra-Cork" strategy described in this article achieves the goal of preserving therapeutic activity while abrogating toxicity. The strategy crucially relies on a judicious matching of the pharmacokinetic properties of the cytokine fusion and the signaling inhibitor.

Toxicity observed with antibody-cytokine fusions, both in preclinical models and in patients, typically emerges shortly after intravenous administration and disappears when the

**Figure 6. Schedule optimization of Ruxolitinib treatment can completely mask "on-target off-tumor" activity of L19-mIL12.**

(A) Quantification of L19-mIL12 levels in plasma at different time points after a single injection in MC-38 tumor-bearing mice (75 mg/kg, s.c.). Results are expressed as ng/mL of antibody-cytokine fusion protein (mean ± SD, $n = 3$ mice per time point). (B,C) Tumor growth curves (B) of MC-38 tumor-bearing mice and treatment tolerability assessed by body weight change (C). Mice received L19-mIL12 alone (1.2 mg/kg, i.v, black arrows) or in combination with Ruxolitinib (75 mg/kg, s.c, gray arrowheads) 10 min before and 6 h after the immunocytokine. Data represent mean tumor volume and body weight change (%) ± SEM ($n = 5-7$ mice per group). Two-way ANOVA analysis (***$p < 0.001$; ****$p < 0.0001$). (D) MC-38 tumor-bearing mice were euthanized at different time points after a single injection of L19-mIL12 (1.2 mg/kg, i.v, black arrow) alone or in combination with Ruxolitinib (75 mg/kg, s.c, gray arrowhead) ($n = 3$ mice per group). Plasma concentration of IFN-γ levels over time. (E) Spleens from each treatment were weighted 24 h after a full therapy cycle ($n = 3$ mice per group). (F,G) Quantification of liver coagulation necrosis 24 h after a full therapy cycle (F) ($n = 3$ mice per group). H&E staining of liver sections (G) at 10× magnification (scale bars = 500 μm; upper panels) and 20× magnification (scale bars = 100 μm; lower panels). Necrotic lesions are limited by the dotted square. Data information: in (D–F), data represent mean ± SD. One-way ANOVA analysis (**$p < 0.01$; ***$p < 0.001$; ****$p < 0.0001$). Source data are available online for this figure.

biopharmaceutical concentration in blood drops below a certain concentration threshold. Adverse events include hypotension (a consequence of vasoactive properties of pro-inflammatory cytokines), nausea and vomiting (which can be controlled with anti-emetic medications), as well as flu-like symptoms and shivers (Johannsen et al, 2010; Spitaleri et al, 2013; Danielli et al, 2015; Weiss et al, 2020; Look et al, 2023), controllable only to some extent with anti-pyretic medications. In the case of IL12, which has long been considered one of the most important cytokine mediators of anti-tumor activity, liver toxicity also needs to be managed (Jia et al, 2022). In a recent study, macrophages and neutrophils have been identified as mediators and effectors of aberrant inflammation driving liver damage in $T_H1$-promoting immunotherapies (Siwicki et al, 2021). These undesired toxicities should be adequately controlled by selective signaling inhibitors such as Ruxolitinib. Here, we have demonstrated that the co-administration of L19-IL12 plus Ruxolitinib, with a judicious schedule, allowed to retain full therapeutic activity while avoiding the onset of liver necrosis at ultra-high doses.

One of the most rewarding findings associated with the use of the "Intra-Cork" technology in mouse models of cancer was the observation that the cytokine payload, kept for long periods of time at the tumor site, is the mediator of therapeutic activity, while cytokine activity in the periphery can effectively be blocked. Tumor resident CD8$^+$ T cells and NK cells are the main mediators of anti-cancer activity in the mouse, as shown by in vivo depletion studies (Puca et al, 2020). Both cell types can be efficiently activated by IL12, anchored at the neoplastic site by means of high-affinity antibodies directed against suitable extracellular matrix components.

Ruxolitinib has also found clinical application as a prophylactic treatment for the management of adverse events associated with bispecific antibodies and CAR-T cell therapies (Uy et al, 2020; Wei et al, 2020; Pan et al, 2021; Zi et al, 2021). The use of suitably spaced administrations of pathway-specific signaling inhibitors represents a general avenue to maximize the clinical utility of antibody cytokine fusions with a long residence time at the neoplastic site. Clinical trials of Ruxolitinib in combination with L19-IL12 in cancer patients are warranted.

# Methods

## Cell lines

MC-38 murine colon adenocarcinoma cells (CVCL_B288; ATCC) were expanded in DMEM medium (Gibco Life Technologies) supplemented with penicillin-streptomycin (Thermo Fisher) and 10% Fetal Bovine Serum (FBS; Gibco Life Technologies). Chinese hamster ovarian cells (CHO, CVCL_0213; ATCC) were cultured in PowerCHO media (Lonza) according to the supplier's instructions. A patient-derived NK cell line (NK-92, CVCL_2142) was kindly provided by Professor C. Münz (University of Zurich, Switzerland) and was cultured in RPMI medium (Gibco Life Technologies) supplemented with penicillin-streptomycin (Thermo Fisher), 10% FBS and recombinant human IL2 (100 U/mL) (Peprotech). Authentication of the cell lines, including the evaluation of post-freeze viability, growth properties, morphology, test for mycoplasma contamination, isoenzyme assay, and sterility tests, was performed by the cell bank before shipment.

## Protein production and characterization

L19-mIL12 is a fusion protein consisting of the L19 antibody in tandem diabody format fused to a single chain of the murine IL12 (comprising p40 and p35 chains) at the N-terminus. The cloning of the protein has been previously described (Puca et al, 2020). The fusion protein was produced in CHO mammalian cells using the stable cell line kindly provided by Dr. Emanuele Puca (Philochem AG). The product was purified from the cell culture medium by protein A affinity chromatography and size exclusion chromatography (SEC) on a Superdex 200 increase 10/300 GL (GE Healthcare) to obtain a monomeric fraction. The clinical-stage product L19-IL12, carrying the human payload, was obtained from Philogen S.p.a. The quality of the proteins was assessed by SDS-PAGE and by SEC. Aminoacidic sequences are reported in the Appendix Tables S1 and S2.

## In vitro IFN-γ release assay

Human NK-92 cells were analyzed for their ability to produce IFN-γ upon incubation with different small-molecule inhibitors. Baricitinib (HY-15315), Tofacitinib (HY-40354), Upadacitinib (HY-19569), and Ritlecitinib (HY-100754) were purchased from MedChemExpress LLC. Ruxolitinib (S1378) and Dexamethasone (S1322) were purchased from Selleckchem. NK-92 cells were initially starved for 4 h in plain RPMI medium. Cells were then resuspended at a density of $1 \times 10^6$ cells/mL in complete RPMI, and 100 μL of the cell suspension was incubated with increasing concentrations of the inhibitors (up to 3 μM) in the presence of L19-IL12 (10 ng/mL). After 24 h, cell-free supernatants were collected, and the concentration of IFN-γ was measured using the ELISA MAX™ Set Human IFN-γ kit (BioLegend).

## Pharmacokinetic profile of L19-mIL12 by ELISA

To assess the pharmacokinetics of L19-mIL12 (1.2 mg/kg) in blood, mice bearing MC-38 tumors were injected intravenously and euthanized at different time points ($n = 3$/time point). Fresh blood was collected in heparin tubes (BD Microtainer LH tubes) and centrifuged. Plasma was frozen and stored at $-80\,°C$. The concentration of L19-mIL12 was measured by ELISA. 96 well plates (MaxiSorp, Sigma) were coated with EDB (100 nM), followed by incubation with plasma samples for 2 h. L19-mIL12 was detected with anti-mouse IL12 (p70) antibody (1:200, clone C18.2, Biolegend) followed by anti-rat-IgG HRP (1:1000, Sigma). Analysis for each condition was carried out with $n = 3$.

## Biodistribution of Ruxolitinib by mass spectrometry

To assess the biodistribution of Ruxolitinib (75 mg/kg), mice bearing MC-38 tumors were injected subcutaneously with Ruxolitinib, followed by an intravenous injection of L19-mIL12 (1.2 mg/kg) 10 min later ($n = 3$/time point). Oral gavage was avoided to comply with the animal welfare regulations of Kanton Zürich. The animals were euthanized at different time points, and organs were excised, frozen, and stored at $-80\,°C$. Fresh blood was collected in heparin tubes (BD Microtainer LH Tubes) and centrifuged. Plasma was frozen and stored at $-80\,°C$. For the mass spectrometry (MS) analysis, tissues were weighted (20 mg) and resuspended in a solution containing 95% ACN, 0.1% formic acid, and the internal standard solution (18 $\mu$M $^2$H$_8$-Ruxolitinib, LGC standards) was added. The samples were homogenized with a tissue lyser (TissueLyser II, QIAGEN) for 15 min at 30 Hz and centrifugated ($1500 \times g$, 10 min). Supernatants were dried at room temperature with a vacuum centrifuge (Eppendorf). The pellets were resuspended in 1 mL of an aqueous solution A (containing 3% ACN and 0.1% of TFA) and subsequently purified on Oasis HLB SPE columns (1 mL volume, 30 mg sorbent, Waters). SPE columns were activated with ACN, equilibrated with solution A, samples were loaded and washed with solution A, and finally, elution was achieved with an aqueous solution B (containing 60% ACN and 0.1% TFA). The eluates were dried under vacuum at room temperature, resuspended in 1 mL of solution A, and further purified on Sep-Pak C18 columns (1 mL volume, 50 mg sorbent, Waters). Columns were activated with ACN, equilibrated with solution A, the samples were loaded and washed with solution A and finally eluted with solution B. Eluates were dried under vacuum at room temperature and resuspended in 900 $\mu$L of an aqueous solution containing 3% ACN and 0.1% of FA. Each sample (1 $\mu$L) was then injected into the LC-MS system. Chromatographic separation was carried out on an Acclaim PepMap RSLC column (50 $\mu$m × 15 cm, a particle size of 2 $\mu$m, a pore size of 100 Å, Thermo Fisher Scientific) with a gradient program from 95% A (H$_2$O, 0.1% FA), 5% B (ACN 0.1% FA) to 5% A and 95% B in 45 min on an Easy nanoLC 1000 (Thermo Fisher Scientific). The flow rate was 300 nL/min, and the column was at room temperature. The LC system was coupled to a Q-Exactive mass spectrometer (Thermo Fisher Scientific) via a Nano Flex ion source (Thermo Fisher Scientific). Ionization was carried out in positive ion mode with a spray voltage of 2 kV, a capillary temperature of 250 °C, and an RF level of 60 S-lens. The mass spectrometer was operated in Single Ion Monitoring (SIM) mode, setting the

following parameters: mass range 716.5113–730.5113 $m/z$, resolution of 70,000 (FWHM at 200 $m/z$), an AGC target of $5 \times 10^4$, and a maximum injection time of 200 ms. Peak areas of analytes and internal standards were integrated, and corresponding ratios were calculated. The ratios were then transformed into pmol/g of wet tissue using single-concentration external calibration points (Appendix Table S3) and corrected by the total weight of the sample analyzed. All biodistribution experiments were performed using three mice per experimental condition. Graphs express mean ± standard deviation values. Data analysis was performed with Thermo Xcalibur Qual Browser v2.2 (Thermo Fisher Scientific) and Prism8 (GraphPad).

## Therapy studies

Seven to 8-week-old C57BL/6J mice (Janvier Labs) were housed in a pathogen-free facility under a controlled 12-h photoperiod, and water and food were provided ad libitum. Female mice were of preference due to more tolerant behavior when kept as a same-sex group. Gender variabilities have not been previously observed for the pharmacokinetic and pharmacodynamic properties of Ruxolitinib (Drenberg et al, 2019) or L19-mIL12 (Weiss et al, 2020). Mice were subcutaneously injected with $1 \times 10^6$ of MC-38 cells in the right flank. The choice of the mouse strain and the tumor model was based on (i) dose-dependent toxicity and (ii) potent anti-tumor activity of L19-mIL12 treatment, which was required to explore the Intra-Cork technology. Tumor volume was measured daily with a caliper and calculated using the formula: volume (mm$^3$) = length × width$^2$ × 0.5. When tumors reached a volume of approximately 100 mm$^3$, mice were randomized in homogeneous groups based on tumor volumes and body weight and received three treatment injections (every 72 h), corresponding to a full therapy cycle (treatment not blinded). L19-mIL12 (0.6 mg/kg, 0.9 mg/kg, 1.2 mg/kg, or 2.4 mg/kg) was injected intravenously into the tail, and Ruxolitinib (75 mg/kg) was given subcutaneously. In the combination group, Ruxolitinib was given 10 min before the L19-mIL12 administration. In selected experiments, Ruxolitinib was given 10 min before and 6 h after L19-mIL12 (1.2 mg/kg) administration. Animals were euthanized when tumors reached a maximum volume of 1500 mm$^3$ or weight loss exceeded 15% from the starting treatment day. Therapy experiments were performed using 5–7 mice per treatment group. Complete statistical analyses are shown in the Appendix Table S4.

## Hematology and liver enzymes analyses

For hematology testing, mice were euthanized 24 h after the last injection and EDTA-anticoagulated blood was immediately assayed with a Sysmex XT-2000iV hematology analyzer (Veterinary Laboratory, University of Zurich, Switzerland). Plasma was obtained by spinning heparin-anticoagulated blood at $1500 \times g$ for 15 min. Liver enzyme levels, including ALT and AST, were quantified using a Roche Integra 800 analyzer (Vetsuisse Faculty, University of Zurich, Switzerland).

## Cytokine quantification in plasma and tumor

For quantification of IFN-$\gamma$ levels in plasma after a single injection of L19-mIL12 (1.2 mg/kg) as monotherapy or in combination with

Ruxolitinib (75 mg/kg), mice were euthanized at different time points. Blood was transferred to heparin tubes (BD Microtainer Tube) and plasma was collected after centrifugation. IFN-γ levels were determined using the ELISA MAX™ Set Murine IFN-γ kit (BioLegend). Quantification of cytokine levels in the tumor and in blood was performed 24 h after the last injection. Blood was transferred to heparin tubes and plasma was collected after centrifugation. Excised MC-38 tumors were treated with a digestion buffer (Tris-HCl 50 mM, pH 7.4, NaCl 0.6 M, Triton X-100 0.2%, BSA 0.5%) and freshly dissolved protease inhibitors (1 mM benzamidine, 0.1 mM benzethonium chloride and 0.1 mM phenylmethylsulfonyl fluoride; Roche). Samples were homogenized with a QIAGEN Tissue Lyser. The levels of IFN-γ, TNF-α, IL10, IL6, and IP10 were measured using an electrochemiluminescence-based assay manufactured by Meso Scale Discovery (MSD, Gaithersburg, MD).

## Immunofluorescence studies

For immunofluorescence analysis, tumors were embedded in a cryo-embedding medium (NEG-50, Thermo Fisher). Cryostat sections (8–10 µm) were stained using the following primary antibodies: goat anti-CD31 (1:200, R&D Systems; AF3628), rabbit anti-CD4 (1:200, Sino Biological, Wayne, PA; 50,134-R001), rabbit anti-Foxp3 (1:200, Invitrogen; 7000914), rabbit anti-NKp46 (1:200, BioLegend; 137602), and rabbit anti-CD8 (1:200, Sino Biological; 50389-R208). Primary antibodies were detected with anti-rabbit AlexaFluor488 (1:200, Invitrogen; A11008) and anti-goat AlexaFluor594 (1:200, Invitrogen; A21209). Cell nuclei were stained using 4′,6-diamidino-2-phenylindole (DAPI) (1:500, Invitrogen; D1306). Slides were mounted with mounting medium (Dako Agilent, Carpinteria, CA) and images were obtained with a Leica DMI6000B confocal microscope (Scientific Center for Optical and Electron Microscopy ScopeM, ETH, Switzerland). Images were analyzed using ImageJ software.

## Flow cytometry studies

For immune cell infiltrates analysis and cytokine quantification by flow cytometry, MC-38 tumor-bearing mice were injected according to the therapy schedule and euthanized 24 h after the last injection. Blood and mechanically dissociated spleens were treated with Red Blood Cell Lysis Buffer (BioLegend) following the manufacturer's protocol. Tumors were treated with an enzyme mix consisting of 1 mg/mL collagenase II (Gibco Life Technologies) and 0.1 mg/mL DNAse I (Roche) in RPMI medium for 60 min at 37 °C. After enzymatic digestion, tumors were filtered through a 70-µm cell strainer. For the cytokine analysis in the spleens and tumors, cells were stimulated with PMA/Ionomycin Cell Activation Cocktail (1:1000, BioLegend) for 4 h at 37 °C, of which the last 2 h were in the presence of Brefeldin-A (1:1000, BioLegend). Discrimination of living cells was performed with Zombie Fixable Viability Kit (BioLegend). Samples were incubated with TruStain FcX™ (BioLegend) to eliminate unspecific signals mediated by Fc receptor binding. Surface staining was performed using the following antibodies (all purchased from BioLegend): anti-CD45-FITC (1:100, clone 30-F11), anti-CD8-PerCP (1:400, clone 53-6.7), anti-CD4-PeCy7 (1:400, clone RM4-5), anti-CD3-BV510 (1:100, clone 17A2), anti-NK1.1-PE (1:200, clone PK136), anti-CD44-

APCCy7 (1:400, clone IM7) and anti-CD62L-BV421 (1:100, clone MEL-14). For intracellular cytokine detection, cells were treated with Fixation Buffer and Permeabilization Buffer (BioLegend) according to the manufacturer's instructions, prior to incubation with anti-IFNγ-APC (1:400, clone XMG1.2). For analysis of STAT4 phosphorylation, mice received a single injection of L19-mIL12 (1.2 mg/kg) as monotherapy or in combination with Ruxolitinib. Mice were euthanized at different time points, and blood was transferred to heparin tubes. Samples were lysed and fixed immediately to preserve phosphorylation events using RBC Lysis/ Fixation Solution (BioLegend) for 15 min at 37 °C and then permeabilized with True-Phos™ Perm Buffer (BioLegend) for at least 1 h at −20 °C. Cells were stained with anti-pSTAT4-APC (1:100, clone 4LURPIE; Invitrogen). The acquisition of samples was performed using a BD Fortessa Flow cytometer and the DIVA software or a CytoFLEX Flow cytometer and the CytExpert software. Data were analyzed using FlowJo software (v10.8.1; BD Biosciences). Representative gating strategies are shown in the Appendix Figs. S1–3.

## Proteomic analysis of livers and tumors

For proteomic analysis, mice were euthanized 24 h after the last injection. Livers and tumors were snap-frozen and stored at −80 °C. Weighted tissues (20 mg) were resuspended in a lysis buffer (50 mM Tris-HCl, 100 mM NaCl, 8 M Urea, pH 8) containing protease inhibitors (Roche). Samples were homogenized using a tissue lyser (TissueLyser II, QIAGEN) and by sonication. Samples were centrifugated for 10 min at $21,000 \times g$, supernatant recovered, and protein concentration measured with a BCA kit (Thermo Fisher). For each sample, 20 µg of protein was diluted with digestion buffer (50 mM Tris-HCl, 1 mM CaCl2, pH 8.0) to a final concentration of 100 µg/mL. Proteins were reduced with TCEP and alkylated with iodoacetamide prior to overnight digestion with trypsin (enzyme-protein mg ratio 1:50) at 37 °C. The resulting tryptic peptides were subjected to $C_{18}$ purification and desalting (Macro Spin Columns, Harvard Apparatus). Peptides were analyzed with an Orbitrap Q-Exactive mass spectrometer coupled to an EASY nanoLC 1000 system via a Nano Flex ion source. Chromatographic separation was carried out at room temperature on an Acclaim PepMap RSLC column (50 µm × 15 cm, particle size 2 µm, pore size 100 Å), using 120 min linear gradient with 5–35% solvent B (0.1% formic acid in acetonitrile) at a flow rate of 300 nL/ min. Ionization was carried out in positive ion mode, with 2 kV of spray voltage, 250 °C of capillary temperature, 60 S-lens RF level. The mass spectrometer was working in a data-dependent mode. MS1 scan range was set from 350 to 1650 $m/z$, the 10 most abundant peptides were subjected to HCD fragmentation with an NCE of 25. Dynamic exclusion was set at 20 s. Raw files were processed with Proteome Discoverer 2.5 (Thermo Fisher) for quantitative analysis. Database searches were performed against the Mus Musculus reference proteome using Sequest as a search engine, Carbamidomethylation of cysteines was set as a fixed modification, oxidation of methionine as variable modification, and trypsin was set as cleavage specificity allowing a maximum of 2 missed cleavages. An intensity-based rescoring of PSM was carried out with Inferys. Data filtering was performed using a percolator with a 1% False Discovery Rate (FDR). The analysis output was exported and further processed with Python, R, and Prism

**The paper explained**

**Problem**

Cytokine-based therapeutics have shown effectiveness in achieving objective responses in specific tumor types. However, their application is hindered by a lack of specificity, leading to undesired effects on healthy tissues, and limiting the ability to escalate therapeutic doses for optimal efficacy. The use of tumor-homing antibodies for cytokine delivery (also known as "immunocytokines") has been shown to significantly increase the therapeutic index of the corresponding payload but still suffers from side effects associated with peak concentrations of the product in blood upon intravenous administration.

**Results**

In this study, we describe and validate a novel strategy to generate immunocytokines with activity-on-demand, acting exclusively at the neoplastic site, sparing healthy tissues. By pre-treating tumor-bearing mice with a JAK2 inhibitor, we were able to transiently mask the systemic toxicity of a tumor-targeted IL12 without compromising the anti-cancer activity. The combinatorial treatment allowed to safely administer higher doses of the immunocytokine that otherwise would have been very toxic (observed by body weight loss, massive systemic cytokine release, and liver damage). We also prove that the schedule optimization of the inhibitor to pharmacokinetically match the immunocytokine peak concentration in blood is crucial to fully control the "on-target-off-tumor" activity of the immunocytokine.

**Impact**

The findings described in this paper may be clinically relevant as cancer patients receiving treatments based on antibody-cytokine biopharmaceuticals, could benefit from higher efficacious doses. Combination regimens with fast-cleared signaling inhibitors promise to transiently control the cytokine signaling in blood and healthy tissues shortly after systemic administration, thus reducing toxicity. Anti-cancer activity of the payload within the tumor mass is less likely to be affected by this strategy as targeted cytokines have been proven to preferentially accumulate at the malignant site for longer period of times.

(GraphPad). Briefly, protein intensities were $\log_2$ transformed, missing values were imputed with the minimum value observed among all samples and intensities were normalized using median normalization. For all conditions, the mean and standard deviation for each protein were calculated, multiple welch t-test was performed, corrected for multiple comparisons (using the Benjiamini–Hochberg correction with 1% FDR), and fold change analyses were carried out between each treatment condition and the saline. Proteins with a $-\log$ (q-value) above 2 and a $\log_2$(fold change) above 1 or below $-1$ were respectively considered up- or down-regulated. Principal Component Analysis (PCA) was performed with Past4 software (Hammer, 2001). Gene Ontology analysis was performed with Metascape (Zhou et al, 2019). Analysis for each condition was carried out with $n = 6$ (3 biological replicates and 2 experimental replicates for biological replicates).

## Histopathological evaluation

For histopathological analysis, mice were euthanized 24 h after the last treatment injection and fixed in 4% neutral-buffered formalin (Formafix). An initial macroscopic examination of each animal included: the external surface of the body, all orifices, the cranial, thoracic, abdominal and pelvic cavities, and their contents, organs, and tissues. Selected tissues were embedded in paraffin and sectioned (5–10 µm). Hematoxylin and eosin (HE) staining and complete pathological evaluation were performed for histopathological findings (Laboratory for Animal Model Pathology, University of Zurich, Switzerland).

## Immunohistochemical evaluation

For immunohistochemistry analysis, mice were euthanized 24 h after the last treatment injection and livers were fixed in 4% neutral-buffered formalin (Formafix). Immunohistochemistry was performed using the horseradish peroxidase (HRP) method to identify macrophages (Iba1+) and T cells (CD3+). Briefly, after deparaffination, sections underwent antigen retrieval in citrate buffer with a pH 6.0 (Iba1) or Tris/EDTA buffer (pH 9) for 20 min at 98 °C (CD3), followed by incubation with the primary antibodies (Iba1, polyclonal antibody, rabbit anti-human, WAKO, 1:750; CD3, monoclonal antibody rabbit anti-mouse, 1:900, Ventana; both antibodies diluted in dilution buffer, Agilent Dako). This was followed by the blocking of endogenous peroxidase (peroxidase block, Agilent Dako) for 10 min at RT and incubation with the appropriate secondary antibodies/detection systems (Iba1: HRP EnVision+ rabbit, Agilent Dako; CD3: OmniMap anti-rabbit HRP, Ventana), all in an autostainer. Sections were subsequently counterstained with hematoxylin. A lymph node from a normal mouse served as a positive control (Laboratory for Animal Model Pathology, University of Zurich, Switzerland).

## Statistical analyses

GraphPad Prism v.9.0 was used for all statistical analysis. The numbers of animals used are indicated in the corresponding figure legends. The sample size was chosen empirically based on our previous experience in the calculation of experimental variability. Treatment groups were assigned in a randomized fashion but not blinded. No analyzed samples were omitted from the report. The samples were omitted from the analysis if insufficient material was available. Replicates are specified for each experiment in the corresponding figure legend. To compare multiple groups, we used analysis of variance (one-way or two-way ANOVA) followed by Bonferroni's correction. Statistical significance was considered for $p$-values below 0.05. $*p < 0.05$; $**p < 0.01$; $***p < 0.001$; $****p < 0.0001$.

## Ethical statement

Animal experiments were conducted according to the protocols approved by Veterinäramt des Kantons Zürich (license number ZH06/2021) in compliance with the Swiss Animal Protection Act (TSchG) and the Swiss Animal Protection Ordinance (TSchV).

# Data availability

The mass spectrometry proteomics data have been deposited to the ProteomeXchange Consortium via the PRIDE (Perez-Riverol et al, 2022) partner repository with the dataset identifier PXD047339.

## Peer review information

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

## Acknowledgements

The authors thank Prof. Christian Münz and Lukas Egli (Institute of Experimental Immunology, UZH) for sharing the NK-92 cell line. The authors thank ScopeM (ETH Zürich) for the use of the microscopy facility. The authors are extremely grateful to Glenn Dranoff, MD, for his contribution to the study supervision and manuscript revision. This research received no external funding.

## Author contributions

**Giulia Rotta**: Conceptualization; Data curation; Formal analysis; Validation; Investigation; Visualization; Methodology; Writing—original draft; Writing—review and editing. **Ettore Gilardoni**: Resources; Data curation; Formal analysis; Methodology; Writing—review and editing. **Domenico Ravazza**: Data curation; Formal analysis; Methodology; Writing—review and editing. **Jacqueline Mock**: Supervision; Project administration; Writing—review and editing. **Frauke Seehusen**: Data curation; Formal analysis; Methodology; Writing—review and editing. **Abdullah Elsayed**: Resources; Writing—review and editing. **Emanuele Puca**: Conceptualization; Supervision; Writing—original draft; Project administration; Writing—review and editing. **Roberto De Luca**: Resources; Supervision. **Christian Pellegrino**: Resources; Writing—review and editing. **Thomas Look**: Resources; Writing—review and editing. **Tobias Weiss**: Resources; Writing—review and editing. **Markus G Manz**: Resources; Writing—review and editing. **Cornelia Halin**: Resources; Supervision; Writing—review and editing. **Dario Neri**: Conceptualization; Resources; Supervision; Methodology; Writing—original draft; Project administration; Writing—review and editing. **Sheila Dakhel Plaza**: Conceptualization; Resources; Data curation; Formal analysis; Supervision; Validation; Investigation; Visualization; Methodology; Writing—original draft; Project administration; Writing—review and editing.

## Disclosure and competing interests statement

DN is a co-founder and shareholder of Philogen S.p.A. (www.philogen.com), a Swiss-Italian Biotech company that operates in the field of ligand-based pharmacodelivery. GR, EG, DR, JM, AE, EP, RDL, and SDP are employees of Philochem AG, a daughter company of Philogen, acting as the discovery unit of the group. CH is a member of the scientific advisory board of Philogen S.p.A. GR and SDP are inventors on a patent application (Patent number: WO2023131611 A1) filed by Philogen S.p.A covering the technology described in this work. The other authors declare that they have no competing interests.

# Expanded View Figures

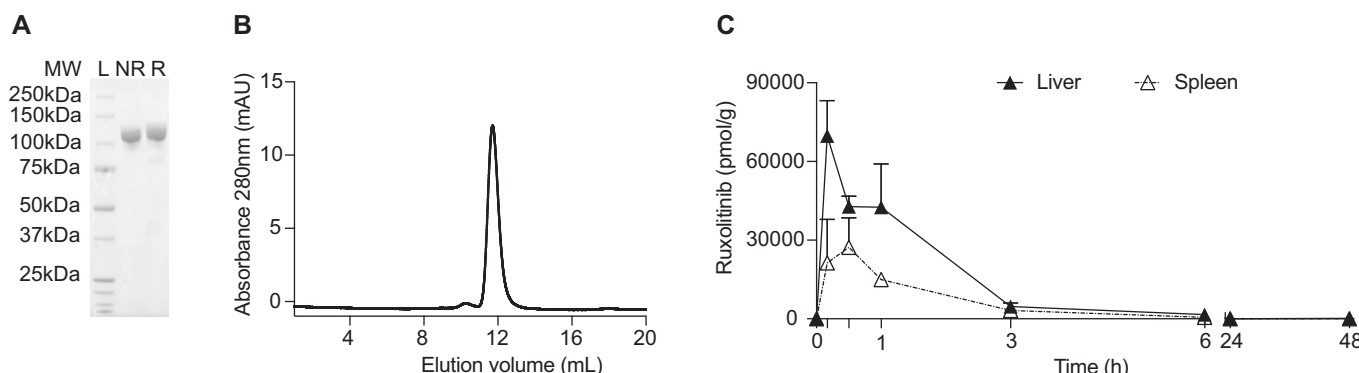

**Figure EV1.  Ruxolitinib exhibits an optimal pharmacokinetic profile with rapid body clearance.**

(A,B) Quality control analysis of L19-IL12 fusion protein assessed by SDS-PAGE (A) and size exclusion chromatography (B). MW = molecular weight; L = ladder; NR = non-reducing conditions; R = reducing conditions. (C) Quantitative ex vivo MS-based biodistribution of Ruxolitinib at different time points after a single dose in MC-38 tumor-bearing mice (75 mg/kg, SC). Results are expressed as picomoles per gram of tissue (mean ± SD, $n = 3$ mice per time point).

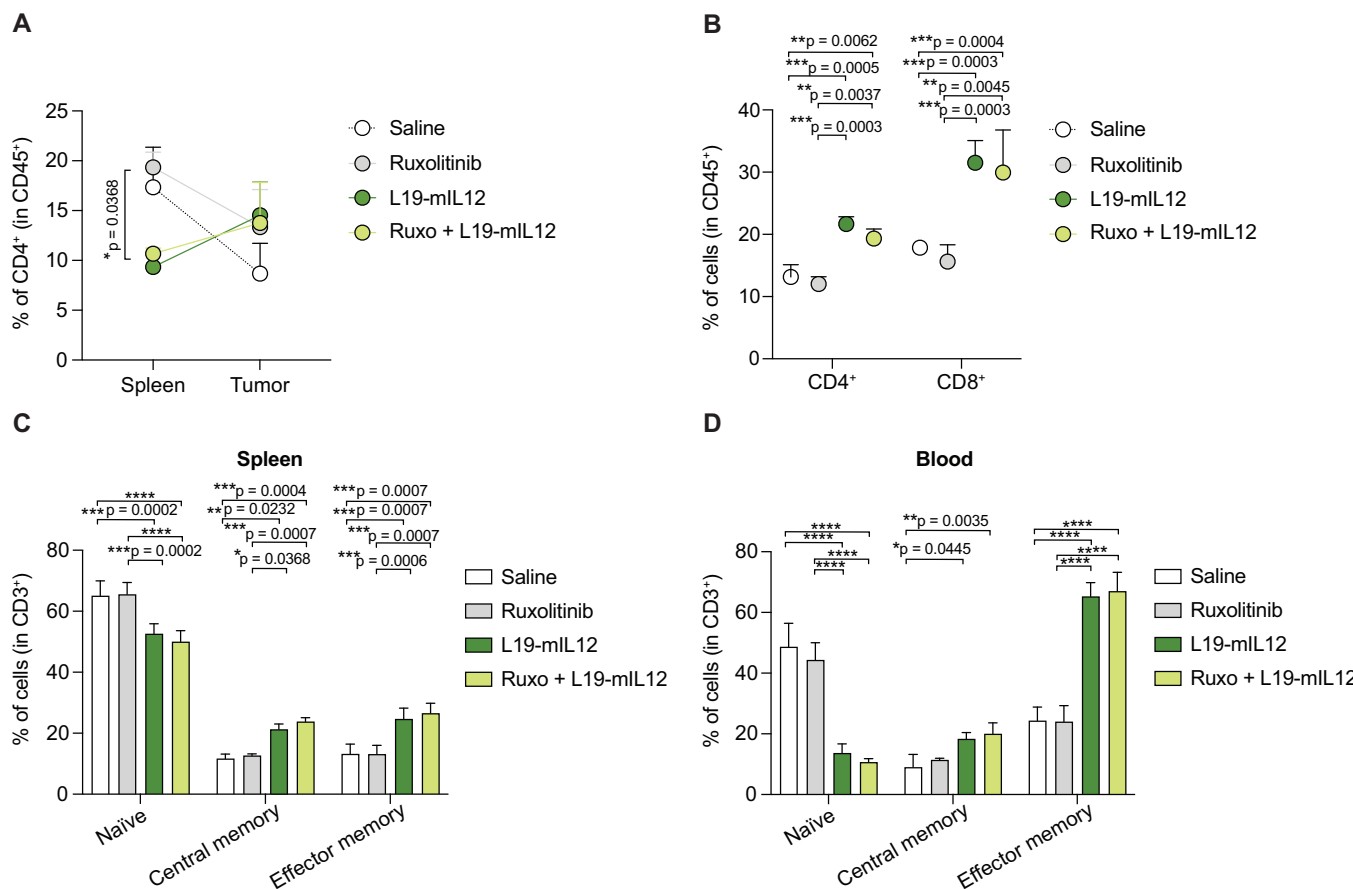

**Figure EV2. Pre-treatment with Ruxolitinib preserves the immunological remodeling induced by L19-mIL12 in blood and spleen.**

MC-38 tumor-bearing mice were euthanized 24 h after the third injection of either saline, Ruxolitinib (75 mg/kg, s.c), L19-mIL12 (1.2 mg/kg, i.v), or Ruxolitinib pre-administered before the L19-mIL12 ($n = 3$–5 mice per group). (A–D) Percentage of CD4 + T cells among CD45+ cells in tumors and spleens (A), percentage of CD8+ and CD4 + T cells among CD45+ cells in blood (B). Percentage of Naïve (CD44$^-$CD62L$^+$), Central Memory (CD44$^+$CD62L$^+$), and Effector Memory (CD44$^+$CD62L$^-$) cells among CD3+ cells in the spleen (C) and blood (D). Data information: in (A–D), data represent mean ± SD. One-way ANOVA analysis (*$p < 0.05$; **$p < 0.01$; ***$p < 0.001$; ****$p < 0.0001$).

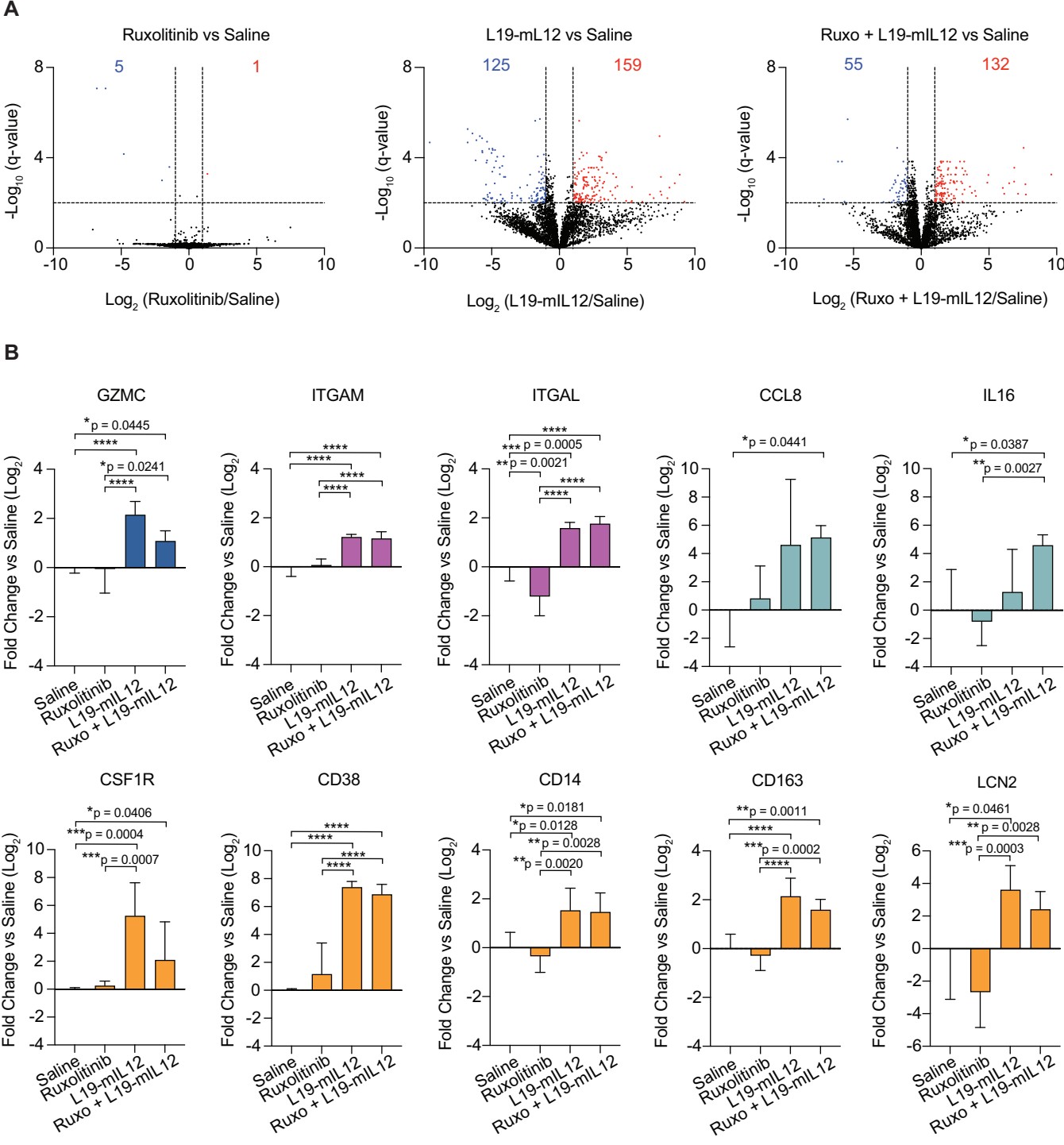

**Figure EV3. Pre-treatment with Ruxolitinib preserves the remodeling of the tumor proteome induced by L19-mIL12.**

MC-38 tumor-bearing mice were euthanized 24 h after the third injection of either saline, Ruxolitinib (75 mg/kg, s.c), or L19-mIL12 (1.2 mg/kg, i.v) alone or in combination with a pre-treatment of Ruxolitinib ($n = 3$ mice per group). (A) Volcano plot representation of the proteomic changes in Ruxolitinib, L19-mIL12 monotherapy or combination, compared to the saline group. Red and blue dots represent significantly up- and down-regulated proteins with FDR < 0.01 and magnitude of change >2-fold. (B) Expression of granzymes (blue), integrins (violet), cytokines (light blue), and immune cell markers (orange) represented as fold change compared to the saline group ($n = 3$ mice per group). Data represent mean ± SD. One-way ANOVA analysis (*$p < 0.05$; **$p < 0.01$; ***$p < 0.001$; ****$p < 0.0001$).

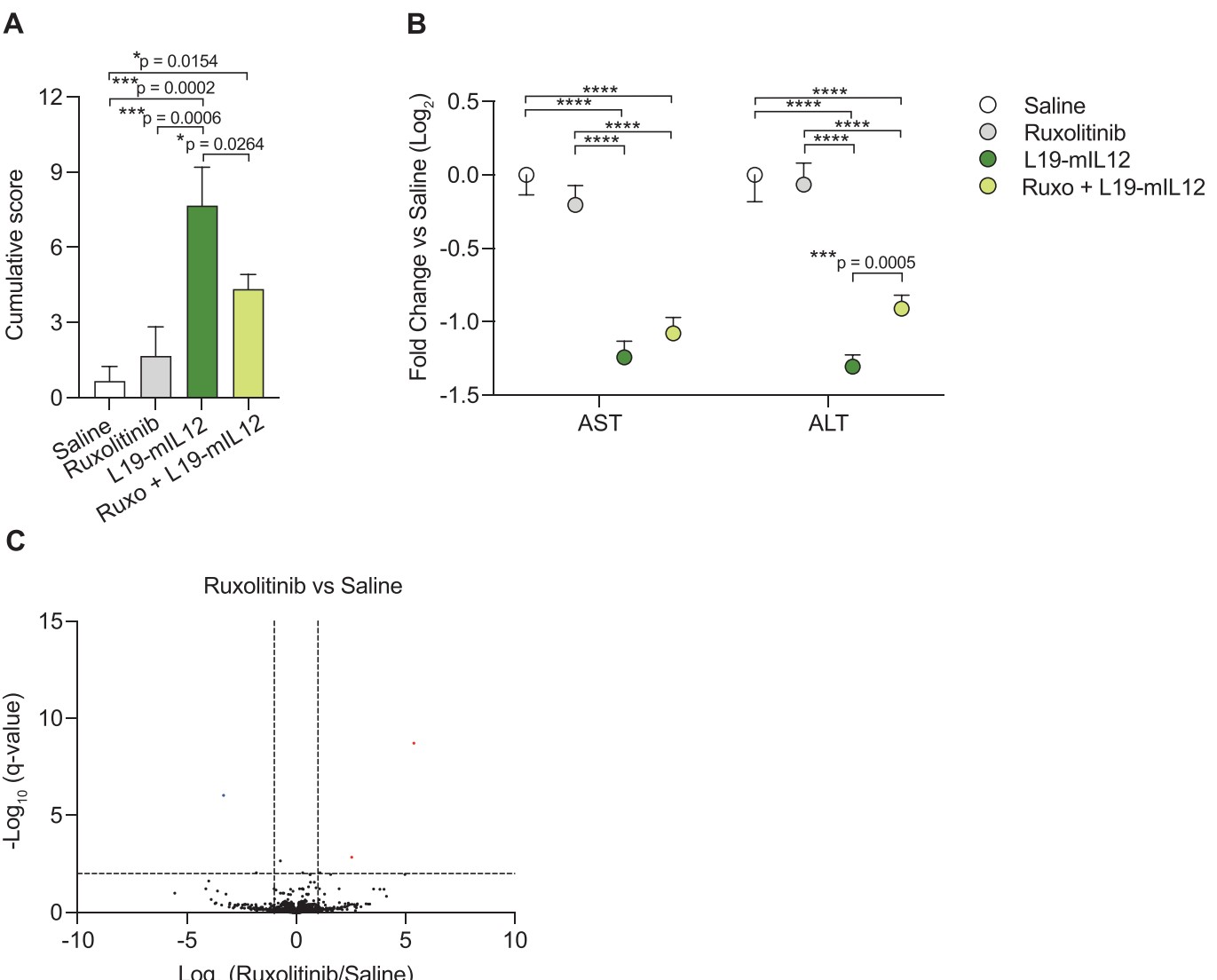

**Figure EV4.  Pre-treatment with Ruxolitinib reduces the hepatic changes associated with high doses of L19-mIL12.**

MC-38 tumor-bearing mice were euthanized 24 h after the third injection of either saline, Ruxolitinib (75 mg/kg, s.c), or L19-mIL12 (1.2 mg/kg, i.v) alone or in combination with a pre-treatment of Ruxolitinib ($n = 3$ mice per group). (**A**) Quantification of liver damage is described as a cumulative score, including coagulation necrosis, single cell necrosis, periportal infiltrates, lobular infiltrates, and vacuolar changes in hepatocytes. (**B**) Levels of aspartate aminotransferase (AST) and alanine aminotransferase (ALT) in liver tissues. (**C**) Volcano plot representation of the liver proteomic changes in the Ruxolitinib group compared to the saline group. Red and blue dots represent significantly up- and down-regulated proteins with FDR < 0.01 and magnitude of change >2-fold. Data information: in (**A,B**), data represent mean ± SD. One-way ANOVA analysis (*$p < 0.05$; ***$p < 0.001$; ****$p < 0.0001$).

**A**

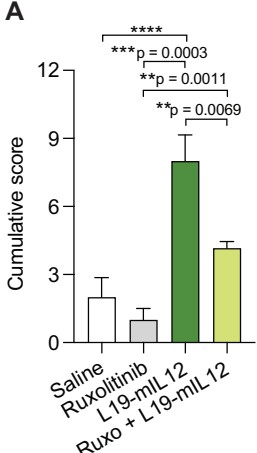

**B**

| Saline | L19-mIL12 | Ruxo + L19-mIL12 |

**C**

| Saline | L19-mIL12 | Ruxo + L19-mIL12 |

◀ **Figure EV5. Schedule optimization completely abrogates the hepatotoxicity associated with high doses of L19-mIL12.**

MC-38 tumor-bearing mice were euthanized 24 h after the third injection of either saline, Ruxolitinib (75 mg/kg, s.c), or L19-mIL12 (1.2 mg/kg, i.v) alone or in combination with Ruxolitinib 10 min before and 6 h after ($n = 3$ mice per group). (**A**) Quantification of liver damage is described as a cumulative score, including coagulation necrosis, single cell necrosis, periportal infiltrates, lobular infiltrates, and vacuolar changes in hepatocytes. Data represent mean ± SD. One-way ANOVA analysis (**$p < 0.01$; ***$p < 0.001$; ****$p < 0.0001$). (**B,C**) Immunohistochemical staining of liver sections for CD3 (**B**) and Iba1 (**C**) at 10× magnification (scale bars = 500 μm; upper panels) and 20× magnification (scale bars = 100 μm; lower panels).

