## [Peer Review File · EMBO Molecular Medicine]

A novel strategy to generate immunocytokines with activity-on-demand using small molecule inhibitors

Giulia Rotta, Ettore Gilardoni, Domenico Ravazza, Jacqueline Mock, Frauke Seehusen, Abdullah Elsayed, Emanuele Puca, Roberto De Luca, Christian Pellegrino, Thomas Look, Tobias Weiss, Markus G. Manz, Cornelia Halin, Dario Neri, and Sheila Dakhel Plaza

Corresponding author: Sheila Dakhel Plaza (sheila.dakhel@philochem.ch) , Dario Neri (dario.neri@philogen.com)

Review Timeline:

Submission Date:	24th Oct 23
Editorial Decision:	17th Nov 23
Revision Received:	18th Dec 23
Editorial Decision:	23rd Jan 24
Revision Received:	29th Jan 24
Accepted:	30th Jan 24

Editor: Lise Roth

Transaction Report:

17th Nov 2023

Dear Dr. Dakhel Plaza,

Thank you for the submission of your manuscript to EMBO Molecular Medicine. We have now received feedback from the three reviewers who agreed to evaluate your manuscript. As you will see from the reports below, the referees acknowledge the interest of the study and are overall supporting publication of your work pending appropriate revisions.

Addressing the reviewers' concerns in full will be necessary for further considering the manuscript in our journal, and acceptance of the manuscript will entail a second round of review.

Please note however that we do not ask you to repeat the experiments in a different mouse strain or sex (comments from referee #1) for this proof-of-concept study, and that while we would welcome the addition of a second mouse model, it will not be mandatory for acceptance of the manuscript.

EMBO Molecular Medicine encourages a single round of revision only and therefore, acceptance or rejection of the manuscript will depend on the completeness of your responses included in the next, final version of the manuscript. For this reason, and to save you from any frustrations in the end, I would strongly advise against returning an incomplete revision.

We are expecting your revised manuscript within three months, if you anticipate any delay, please contact us.

We require:

4) A .docx formatted letter INCLUDING the reviewers' reports and your detailed point-by-point responses to their comments. As part of the EMBO Press transparent editorial process, the point-by-point response is part of the Review Process File (RPF), which will be published alongside your paper.

5) A complete author checklist, which you can download from our author guidelines (<https://www.embopress.org/page/journal/17574684/authorguide#submissionofrevisions>). Please insert information in the checklist that is also reflected in the manuscript. The completed author checklist will also be part of the RPF.

6) Please note that all corresponding authors are required to supply an ORCID ID for their name upon submission of a revised manuscript. An ORCID identifier is currently missing for Prof. Dario Neri.

7) It is mandatory to include a 'Data Availability' section after the Materials and Methods. Before submitting your revision, primary datasets produced in this study need to be deposited in an appropriate public database, and the accession numbers and database listed under 'Data Availability'. Please remember to provide a reviewer password if the datasets are not yet public (see <https://www.embopress.org/page/journal/17574684/authorguide#dataavailability>).

8) For data quantification: please specify the name of the statistical test used to generate error bars and P values, the number (n) of independent experiments (specify technical or biological replicates) underlying each data point and the test used to calculate p-values in each figure legend. The figure legends should contain a basic description of n, P and the test applied. Graphs must include a description of the bars and the error bars (s.d., s.e.m.). Please provide exact p values.

13) Author contributions: CRediT has replaced the traditional author contributions section because it offers a systematic machine readable author contributions format that allows for more effective research assessment. Please remove the Authors Contributions from the manuscript and use the free text boxes beneath each contributing author's name in our system to add specific details on the author's contribution. More information is available in our guide to authors.

16) As part of the EMBO Publications transparent editorial process initiative (see our Editorial at <http://embomolmed.embopress.org/content/2/9/329>), EMBO Molecular Medicine will publish online a Review Process File (RPF) to accompany accepted manuscripts.

In the event of acceptance, this file will be published in conjunction with your paper and will include the anonymous referee reports, your point-by-point response and all pertinent correspondence relating to the manuscript. Let us know whether you agree with the publication of the RPF and as here, if you want to remove or not any figures from it prior to publication. Please note that the Authors checklist will be published at the end of the RPF.

I look forward to receiving your revised manuscript.

Yours sincerely,

Lise Roth
Lise Roth, PhD
Senior Editor
EMBO Molecular Medicine

***** Reviewer's comments *****

Referee #1 (Comments on Novelty/Model System for Author):

While this approach to delivering a pro inflammatory cytokine to a tumor for immunotherapy is novel, translatable, and overall quite exciting, before advancing the idea into the field, this reviewer has a few questions/concerns about the chosen model systems.

1. Why is chosen tumor model MC38? While I acknowledge that this is an extremely common model, it is in a lot of respects very artificial and not terribly physiologically relevant to most human cancers. Can the main findings of this study be repeated in a non-flank tumor model, maybe a spontaneous genetic tumor?
2. Why are only female mice used in this study? Is there sex affect? This choice should be justified.
3. C57Bl6 mice are highly skewed towards TH1 responses, making an IL-12-mediated therapy most likely to succeed in this strain. People, however, will be highly variable in their immune responses. Has this therapy been observed to be successful in other strains that are more TH2 skewed, such as BALBc?

Referee #1 (Remarks for Author):

If the remarks on the models above are not visible to the authors, this reviewer highly recommends repeating key findings of the study (namely reduced tumor volume and reduced body weight loss) in

- a) a different, more physiologically relevant tumor model
- b) both male and female mice
- c) a tumor model in a an additional mouse strain

The translatability of this study might be somewhat limited by not exploring other models, as human beings and the spectrum of cancers they get are highly variable.

Additional minor concerns:

1. The study is prefaced with a screen of different JAK inhibitors using blockage of IFN γ by NK-92 cells as a read out. Based on these results ALONE, the authors choose to proceed with ruxolitinib. Souldn't an in-vivo screen of these inhibitors be performed looking at serum cytokine levels, L19-IL12 levels in relevant tissues, an IFN γ reporter mouse, and/or pSTAT4 staining as a readout? I'm afraid we might be missing the potential benefit of other JAK inhibitors by limiting their exploration to one in-vitro screen on one cell type.
2. Ruxolitinib is orally bioavailable. It is one of the appeals of using it in a translational study. Why, then, is it administered subcutaneously in this study rather than by oral gavage (or maybe a modified chow--although I note the impossibility of controlling for dosing and scheduling with a chow)?
3. In the final figure, could authors please also quantify the L19-mIL12 levels in the tumor following the schedule optimization? Does the repeated dosing of the ruxo alter its availability at the site? And yes, I ask this in spite of the continued observed reduction in tumor volume; thus, it is a minor point.

Referee #2 (Remarks for Author):

In this work, the authors propose a clever strategy to mitigate the systemic toxicity universally associated with intravenously

administered immunocytokines. Such toxicity has been acknowledged by many in the field previously, and motivated such strategies as masking, cis-targeting, and intratumoral administration. In fact, Philogen's Phase III PIVOTAL trial of an L19-IL2 & L19-TNF cocktail utilized intratumoral administration to mitigate toxicity.

The proposed strategy works remarkably well, causing no decrease in efficacy while almost eliminating toxicity. Given the simplicity of the strategy, this represents a tremendous return on invested effort. The authors thoroughly characterize the immunological biomarker correlates of both efficacy and toxicity, and they are convincing.

Novelty of this communication is somewhat diminished by the same organization's publication 4 years ago of essentially the same strategy applied to TNF inhibition and an L19-TNF therapeutic. It would be appropriate to cite their own precedent in the introduction, rather than in the final paragraph as the 69th citation.

It is not clear how coining of the term "Intra-Cork" either advances understanding or has a noncommercial function in a peer-reviewed published article. The authors may want to reconsider the negative effect such gratuitous branding might have on an academic reader.

Referee #3 (Remarks for Author):

In this manuscript, the authors propose the so-called "Intra-Cork" strategy. The strategy consists in combining the administration of the immunocytokine L19-IL12 with a pretreatment of the JAK2 inhibitor Ruxolitinib. According to the concept, the systemic activity of IL-12 is inhibited before antibody-mediated enrichment at the tumor site is achieved. Thus, by leveraging the pharmacokinetic differences of the drugs, systemic toxicity of the cytokine is reduced without compromising antitumor efficacy. This is a clever approach to address the major challenge of systemic toxicity in the application of cytokines in cancer therapy. The authors demonstrate the concept in a syngeneic solid tumor mouse model, employing a mouse surrogate immunocytokine (L19-mIL12). Indeed, the combined treatment here leads to similar tumor growth inhibition but with reduced symptoms of systemic toxicity. Results from treatment schemes with different dosages and frequencies suggest that the system can be adjusted and optimized.

Extensive and detailed analyses of immune cells and tissues from the animal experiments address the impact of Ruxolitinib on the L19-mIL12 effect in the tumor, "on-target off-tumor" activity, and hepatotoxicity.

The manuscript is well-written, with experiments adequately designed and data supporting the conclusions.

Minor comments:

- Quality control (SDS-PAGE and size exclusion chromatography) is shown twice for L19-mIL12 (Fig. 1 and Suppl1), but not for L19-IL12, as indicated in the text.
- Abbreviations should be in capital letters. In line 809, replace "SDS-page" with "SDS-PAGE"; in line 853, replace "Elisa" with "ELISA".

Dr. Sheila Dakhel Plaza
Philochem AG
Libernstrasse 3
CH-8112 Otelfingen
sheila.dakhel@philochem.ch

phone: +41 (78) 404 02 28
<https://www.philochem.ch>

For the attention of

Dr. Lise Roth, Assigned Senior Scientific Editor
c/o *EMBO Molecular Medicine*

And of

Prof. Dr. Philippe Sansonetti, Editor-in-Chief
c/o *EMBO Molecular Medicine*

Institute Pasteur, 25-28 Rue du Dr Roux, 75015 Paris, France

Zürich, 18th December 2023

Re: Resubmission of Manuscript ID: EMM-2023-18880 to EMBO Molecular Medicine

Dear Dr. Roth, Dear Prof. Sansonetti,

Many thanks for your letter of November 17th 2023, containing the Referees' comments to our Manuscript EMM-2023-18880 entitled:

G.Rotta, E.Gilardoni, D.Ravazza, J.Mock, F.Seehusen, A.Elsayed, E.Puca, R.De Luca, C.Pellegrino, T.Look, T.Weiss, MG.Manz, C.Halin, D.Neri & S.Dakhel Plaza:

"A novel strategy to generate immunocytokines with activity-on-demand using small molecule inhibitors"

We now submit a revised version of the manuscript, which addresses the issues raised by the Referees. Our comments are indicated in **red** (see below).

Referees' Comments to Author:

Referee #1 (Comments on Novelty/Model System for Author):

While this approach to delivering a pro-inflammatory cytokine to a tumor for immunotherapy is novel, translatable, and overall quite exciting, before advancing the idea into the field, this reviewer has a few questions/concerns about the chosen model systems.

We are grateful to Referee #1 for his/her comments on our manuscript describing the “Intra-Cork” technology, a new strategy to control side effects associated with tumor-targeted cytokine therapies. Below, we are addressing each of the Referee’s concerns in more detail.

Specific comments:

1.1 Why is chosen tumor model MC-38? While I acknowledge that this is an extremely common model, it is in a lot of respects very artificial and not terribly physiologically relevant to most human cancers.

We appreciate the fair criticism of Referee #1 concerning the choice of the MC-38 tumor model for our *in vivo* experiments.

The main focus of our work was to validate the idea of using pathway-specific small molecule inhibitors of IL12 signaling to control toxicity without impacting efficacy. For that, the possibility of monitoring toxicity induced by IL12 was a crucial requirement to consider when designing our mouse studies. The C57BL/6 is a mouse strain commonly used in cancer research, which shows high sensitivity to IL12-mediated toxicity (1). MC-38 is a model of murine colon adenocarcinoma originally derived from a tumor induced by the chemical carcinogen azoxymethane in the C57BL/6 strain (2).

The possibility of performing our *in vivo* studies in Patient-Derived Xenografts, which have shown superiority in recapitulating human tumor characteristics, is impaired by the lack of an intact immune system of the mice in which these tumors require to be implanted (e.g., SCID mice). As murine IL12 is a strong stimulator of NK and T cells, both components are needed to fully explore the efficacy and safety of L19-mIL12.

The list below outlines all the characteristics needed, which the MC-38 model has, to fully explore the activity and safety of L19-mIL12 in the context of the Intra-Cork technology:

- high toxicity induced by the administration of high dose L19-mIL12 (see explanation in paragraph 1.3)
- expression of the tumor antigen EDB-Fibronectin, the cognate antigen of the L19 antibody in L19-mIL12
- availability of the tumor cell line in-house
- potent anti-tumor activity induced by L19-mIL12 as single agent to assess if the Intra-Cork impairs its activity
- low probability of tumor ulceration (i.e., low severity degree in the mouse, compatible with the requirements of Kanton Zürich, where we perform our *in vivo* research)
- low variability between tumor growth among different mice implanted with subcutaneous tumors (i.e., small standard deviation)
- need of a murine tumor cell line to work with immunocompetent mouse models to fully explore the activity of murine IL12

Can the main findings of this study be repeated in a non-flank tumor model, maybe a spontaneous genetic tumor?

In the past, our group has shown that L19-mIL12 induces a potent therapeutic activity in several preclinical models, including orthotopic (e.g., GL261 glioma, CT2A glioma, KPC06 pancreas) and subcutaneously (e.g., WEHI-164 fibrosarcoma, CT26 colon adenocarcinoma, Lewis Lung Carcinoma, F9 teratocarcinoma) implanted tumors (3,4). Based on the pre-clinical data that we published, Dodekin, a fully human L19-IL12 fusion protein, has been moved to a Phase I clinical trial for the treatment of multiple solid tumor indications.

The objective of this study was to validate the Intra-Cork technology when applied to L19-mIL12. We believe that the findings in the MC-38 model presented in this manuscript (the model selected based on the arguments indicated above) give sufficient motivation to test the Intra-Cork technology in patients treated with Dodekin.

1.2 Why are only female mice used in this study? Is there sex affect? This choice should be justified.

We thank Referee #1 for mentioning his/her concerns regarding the impact of gender on the efficacy of the Intra-Cork approach. No significant impacts of gender in pharmacokinetics or pharmacodynamics have been reported for Ruxolitinib (Intra-Cork inhibitor for IL12) in humans, dogs, rats, and mice (5,6). Previous preclinical studies using L19-mIL12 have been performed in female and male mice, describing no significant differences in the toxicity or therapeutic activity (4). Therefore, we do not anticipate male-to-female variability for the Ruxolitinib / L19-mIL12 Intra-Cork combination.

In general, our *in vivo* studies are biased towards the use of female animals due to the more tolerant behavior when kept as a same-sex group. We normally house 5 female mice per cage to ensure socialization. Laboratory male mice are often housed individually due to aggressive behavior, and according to the Swiss Animal Welfare Ordinance (art. 119), single housing of incompatible animals must be avoided.

Moreover, the MC-38 cell line was generated in female C57BL/6 mice. Thus, we decided to inject the tumors in the strain/sex of origin.

Following Referee #1 suggestion, we have now included in the Material and Methods section a justification for using female mice (Therapy studies: line 445).

- 1.3 C57BL6 mice are highly skewed towards TH1 responses, making an IL-12-mediated therapy most likely to succeed in this strain. People, however, will be highly variable in their immune responses. Has this therapy been observed to be successful in other strains that are more TH2 skewed, such as BALBc?

We thank Referee #1 for his/her comments on the choice of the mouse strain. We agree with the statement on the potential patient-to-patient variability in the clinical efficacy of L19-IL12 and we are aware that immune responses in BALBc and BL6 mice are TH2 and TH1 skewed, respectively. As mentioned in paragraph 1.1., L19-mIL12 was tested in different mouse models, including BALB-c mice bearing CT26 tumors. As evidenced in the graph below, L19-mIL12 shows potent single-agent activity inducing 3/5 complete responses. Hence, the immunocytokine appears to be very active irrespective of the strain background of the mice (3).

Figure for reviewers removed

An explanation of the selection of the MC-38 model in C57BL/6 mice has been added in the revised Material and Methods section (Therapy studies; line 448).

Referee #1 (Remarks for Author):

If the remarks on the models above are not visible to the authors, this reviewer highly recommends repeating key findings of the study (namely reduced tumor volume and reduced body weight loss) in

- a) a different, more physiologically relevant tumor model → please refer to the explanation in point 1.1
- b) both male and female mice → please refer to the explanation in point 1.2
- c) a tumor model in an additional mouse strain → please refer to the explanation in point 1.3

The translatability of this study might be somewhat limited by not exploring other models, as human beings and the spectrum of cancers they get are highly variable.

We hope that our explanations of each of his/her points, together with the changes to the manuscript, satisfy the curiosity and questions of Referee #1.

Additional minor concerns:

- 1.4 The study is prefaced with a screen of different JAK inhibitors using blockage of IFN γ by NK-92 cells as a readout. Based on these results ALONE, the authors choose to proceed with ruxolitinib. Shouldn't an in-vivo screen of these inhibitors be performed by looking at serum cytokine levels, L19-IL12 levels in relevant tissues, an IFN γ reporter mouse, and/or pSTAT4 staining as a readout? I'm afraid we might be missing the potential benefit of other JAK inhibitors by limiting their exploration to one in-vitro screen on one cell type.

We thank Referee #1 for suggesting a screening of potential inhibitors of IL12 signaling activity based on additional *in vivo* studies. We based the choice of Ruxolitinib as an "Intra-Cork" molecule after an *in vitro* screening of different JAK inhibitors. The results, indeed, showed that the most potent modulator of IL12 activation (as measured by IFN- γ release in NK cells) was Ruxolitinib, exhibiting an IC50 of 42nM, significantly superior to Baricitinib (462nM) or Tofacitinob (3'900nM); see main Figure 1D. We anticipate that other potent and selective JAK2 inhibitors (such as Baricitinib or Tofacitinib) could be suited for the devised technology. However, we decided to move to *in vivo* studies with the most potent inhibitor (Ruxolitinib) because the scope of our work was to i) validate the proof-of-concept of using pathway-specific inhibitors of the IL12 signaling cascade and ii) decrease the number of animals used in our preclinical studies (3Rs principle – replace, reduce, refine).

Moreover, an important reason that made us confident in choosing Ruxolitinib, among others, was that the drug had been previously tested clinically to manage treatment-emerging adverse events when receiving other immunotherapies. Here are some reported clinical studies:

- Adjuvant ruxolitinib therapy relieves steroid-refractory cytokine-release syndrome without impairing chimeric antigen receptor-modified T-cell function (9)
- Ruxolitinib mitigates steroid-refractory CRS during CAR T therapy (10)
- Prophylactic Ruxolitinib for Cytokine Release Syndrome (CRS) in Relapse/Refractory (R/R) AML Patients Treated with Flotetuzumab (11)

These references are already cited in the discussion of the revised manuscript.

Based on pre-clinical findings, Ruxolitinib appears to be the ideal candidate to control emerging immune-related adverse events driven by high doses of L19-IL12, as it completely abrogates liver toxicity, cytokine release, body weight loss without impairing the anti-cancer properties.

- 1.5 Ruxolitinib is orally bioavailable. It is one of the appeals of using it in a translational study. Why, then, is it administered subcutaneously in this study rather than by oral gavage (or maybe a modified chow--although I note the impossibility of controlling for dosing and scheduling with a chow)?

We are grateful to the Referee's #1 comment on the route of administration choice.

We fully agree that oral administration of Ruxolitinib would have been the ideal route since it is the same route used in the clinic. The reason we did not do oral gavage was due to our animal license

restrictions. Also, our license does not allow two intravenous injections in the same mouse on the same day.

For these reasons, we decided to inject Ruxolitinib subcutaneously. To validate that the pharmacokinetic properties of the inhibitor after subcutaneous injection were similar to those reported in the literature, we quantified the drug in the blood, tumor, and healthy organs by *ex vivo* MS-based biodistributions (methodology developed in-house). As reported in Figure 1E and supplementary Expanded View Figure 1C, the pharmacokinetics properties (including T1/2 and T_{max}) were comparable to the values reported after oral administration in mice. This gave us the confidence to perform our *in vivo* studies using subcutaneous administration of the inhibitor, thus allowing us to validate the proof-of-concept of the combinatorial treatment.

An extended explanation has been added to the Material and Methods section of the revised manuscript (Biodistribution of Ruxolitinib by Mass Spectrometry; line 437).

- 1.6 In the final figure, could the authors please also quantify the L19-mIL12 levels in the tumor following the schedule optimization? Does the repeated dosing of the ruxo alter its availability at the site? And yes, I ask this in spite of the continued observed reduction in tumor volume; thus, it is a minor point.

We appreciate the suggestion of Referee #1 of quantifying levels of the L19-mIL12 in the tumor after 2x Ruxolitinib administrations. It is important to mention that mice treated with L19-mIL12 at high doses benefit from a significant tumor reduction and, therefore, the material (tumor samples) available for subsequent analyses is very limited.

Biodistribution studies with radiolabeled proteins would be the ideal methodology to confirm that tumor levels of L19-mIL12 were not affected by a double injection of Ruxolitinib. Unfortunately, we did not perform these studies. As Referee #1 points out, the fact that tumor growth was significantly reduced after the schedule optimization of the inhibitor indicates that the tumor-targeting properties of L19-mIL12 are not affected by the treatment with the Intra-Cork compound.

We have added the following sentence to clarify this point in the revised manuscript (Results section; line 307): "The observed tumor growth retardation observed in this study indicates that the tumor-targeting properties of L19-mIL12 are not affected by the treatment with the Intra-Cork inhibitor"

Referee #2 (Remarks for Author):

In this work, the authors propose a clever strategy to mitigate the systemic toxicity universally associated with intravenously administered immunocytokines. Such toxicity has been acknowledged by many in the field previously, and motivated such strategies as masking, cis-targeting, and intratumoral administration. In fact, Philogen's Phase III PIVOTAL trial of an L19-IL2 & L19-TNF cocktail utilized intratumoral administration to mitigate toxicity.

The proposed strategy works remarkably well, causing no decrease in efficacy while almost eliminating toxicity. Given the simplicity of the strategy, this represents a tremendous return on invested effort. The authors thoroughly characterize the immunological biomarker correlates of both efficacy and toxicity, and they are convincing.

We are extremely grateful to Referee #2 for his/her nice comments. He/she highlights the main findings and immediate clinical translation potential of the devised technology.

2.1 Novelty of this communication is somewhat diminished by the same organization's publication 4 years ago of essentially the same strategy applied to TNF inhibition and an L19-TNF therapeutic. It would be appropriate to cite their own precedent in the introduction, rather than in the final paragraph as the 69th citation.

We appreciate the suggestions of Referee #2 to change [69]. We have now changed the text to anticipate our previous work on L19-TNF and RIPK1 inhibitors in the Introduction section (line 144).

2.2 It is not clear how coining of the term "Intra-Cork" either advances understanding or has a noncommercial function in a peer-reviewed published article. The authors may want to reconsider the negative effect such gratuitous branding might have on an academic reader.

We thank Referee #2 for the comment regarding the technology's name.

The "Intra-Cork" term was born with the analogy of a glass bottle, representing the activity of an immunocytokine (e.g., L19-IL12), whose content (the activity of the payload) is contained by a cork (e.g., Ruxolitinib, the small molecule inhibitor). Given the fact that this particular cork acts on the cytokine receptor intracellular signaling pathway, the term "Cork" was paired with the adjective "Intra": Intra-Cork.

In order to make the reader more comfortable with the term and to explain the reasoning behind it, a more detailed explanation has been added to the revised version of the manuscript (Results section; line 157).

Referee #3 (Remarks for Author):

In this manuscript, the authors propose the so-called "Intra-Cork" strategy. The strategy consists in combining the administration of the immunocytokine L19-IL12 with a pretreatment of the JAK2 inhibitor Ruxolitinib. According to the concept, the systemic activity of IL-12 is inhibited before antibody-mediated enrichment at the tumor site is achieved. Thus, by leveraging the pharmacokinetic differences of the drugs, systemic toxicity of the cytokine is reduced without compromising antitumor efficacy. This is a clever approach to address the major challenge of systemic toxicity in the application of cytokines in cancer therapy.

The authors demonstrate the concept in a syngeneic solid tumor mouse model, employing a mouse surrogate immunocytokine (L19-mIL12). Indeed, the combined treatment here leads to similar tumor growth inhibition but with reduced symptoms of systemic toxicity. Results from treatment schemes with different dosages and frequencies suggest that the system can be adjusted and optimized.

Extensive and detailed analyses of immune cells and tissues from the animal experiments address the impact of Ruxolitinib on the L19-mIL12 effect in the tumor, "on-target off-tumor" activity, and hepatotoxicity.

The manuscript is well-written, with experiments adequately designed and data supporting the conclusions.

We thank Referee #2 for the excellent summary of our manuscript. We are pleased to see that he/she appreciates the main idea and findings of this study and that he/she believes that the amount and quality of the experiments are adequate for the publication of our work in *EMBO Molecular Medicine*.

Minor comments:

- 3.1 Quality control (SDS-PAGE and size exclusion chromatography) is shown twice for L19-mIL12 (Fig. 1 and Suppl1), but not for L19-IL12, as indicated in the text.

We thank Referee #3 for noticing this. We had erroneously named the figure legend in the SUPPLEMENTARY FIGURES with L19-mIL12 twice. We have corrected the mistake in Figure EV1, which shows the SDS-PAGE and size exclusion chromatography of L19-IL12 (clinical-stage product carrying the human payload).

- 3.2 Abbreviations should be in capital letters. In line 809, replace "SDS-page" with "SDS-PAGE"; in line 853, replace "Elisa" with "ELISA".

We thank Referee #3 for his/her corrections. We have now replaced "SDS-page" with "SDS-PAGE" and "Elisa" with "ELISA."

We thank you for your consideration in this matter and hope that the paper may be now acceptable for publication.

Best regards,

Dr. Sheila Dakhel Plaza

Prof. Dr. Dario Neri

1. Nakamura S, Otani T, Ijiri Y, Motoda R, Kurimoto M, Orita K. IFN- γ -Dependent and -Independent Mechanisms in Adverse Effects Caused by Concomitant Administration of IL-18 and IL-12. *J Immunol.* 2000;164.
2. Corbett TH, Griswold DP, Roberts BJ, Peckham JC, Schabel FM. Tumor Induction Relationships in Development of Transplantable Cancers of the Colon in Mice for Chemotherapy Assays, with a Note on Carcinogen Structure. *Cancer Res.* 1975;35.
3. Puca E, Probst P, Stringhini M, Murer P, Pellegrini G, Cazzamalli S, et al. The antibody-based delivery of interleukin-12 to solid tumors boosts NK and CD8+ T cell activity and synergizes with immune checkpoint inhibitors. *Int J Cancer.* 2020;146.
4. Weiss T, Puca E, Silginer M, Hemmerle T, Pazahr S, Bink A, et al. Immunocytokines are a promising immunotherapeutic approach against glioblastoma. *Sci Transl Med.* 2020;12.
5. EMA. Jakavi assessment report. *Ema.* 2011;44.
6. Drenberg CD, Shelat A, Dang J, Cotton A, Orwick SJ, Li M, et al. A high-throughput screen indicates gemcitabine and JAK inhibitors may be useful for treating pediatric AML. *Nat Commun.* 2019;10.
7. Leonard JP, Sherman ML, Fisher GL, Buchanan LJ, Larsen G, Atkins MB, et al. Effects of single-dose interleukin-12 exposure on interleukin-12 associated toxicity and interferon- γ production. *Blood.* 1997;90.
8. Yeleswaram S, Smith P, Burn T, Covington M, Juvekar A, Li Y, et al. Inhibition of cytokine signaling by ruxolitinib and implications for COVID-19 treatment. *Clin. Immunol.* 2020.
9. Wei S, Gu R, Xu Y, Liu X, Xing Y, Gong X, et al. Adjuvant ruxolitinib therapy relieves steroid-refractory cytokine-release syndrome without impairing chimeric antigen receptor-modified T-cell function. *Immunotherapy.* 2020;12.
10. Pan J, Deng B, Ling Z, Song W, Xu J, Duan J, et al. Ruxolitinib mitigates steroid-refractory CRS during CAR T therapy. *J Cell Mol Med.* 2021;25.
11. Uy GL, Rettig MP, Christ S, Aldoss I, Byrne MT, Erba HP, et al. Prophylactic Ruxolitinib for Cytokine Release Syndrome (CRS) in Relapse/Refractory (R/R) AML Patients Treated with Flotetuzumab. *Blood.* 2020;136.

23rd Jan 2024

Dear Dr. Dakhel Plaza,

Thank you for submitting your revised manuscript, and please accept my apologies for the delay in getting back to you in this busy time of the year. We have now received the report from the referee who re-reviewed your manuscript. As you will see below, this referee is satisfied with the revisions, and I will therefore be able to accept your manuscript once the following editorial points will be addressed:

1/ Manuscript text:

- Please remove the yellow highlights, accept the previous changes, and only keep in track changes mode any new modification.
- Materials and Methods:
 - o Please add the Supplementary Methods, currently in the Appendix, to the main manuscript file (we do not have size restriction).
 - o Please provide the antibody dilutions/concentrations for each experiment.
 - o Statistics: please include a statement on blinding, randomization, inclusion/exclusion criteria, sample size, and correct the checklist accordingly.
- Data availability: Thank you for depositing your data in an external repository. Please note that the datasets must be publicly available before acceptance.

2/ Figures and Appendix:

- Please provide exact p values, not a range, in the figures or their legends.
- Please add highlight boxes to figure EV5B and C.
- Appendix: please add legends to the Appendix figures. Methods should be included in the main manuscript text. Table S1 should be renamed Appendix Table S1.
- Figure Legends:
 - Please note that the error bar definition related information in the legend of figure 2a-h is incorrectly labelled as 2a-g. This needs to be rectified.
 - Please note that in figures 5b-c; 6b-f; EV 4a-b; EV 5a; there is a mismatch between the annotated p values in the figure legend and the annotated p values in the figure file that should be corrected.
 - Please note that for the figures 5d-f, p-values and statistical tests are indicated in the legends. However, comparison for the same, ""*****/**/*/*"" has not been represented in the figures. Please rectify this in the figures or legends as applicable.
 - Please note that the measure of center for the error bars needs to be defined in the legends of figure 1e; EV 1c; EV 3b.

3/ Checklist:

- Please complete the Statistic section.
- Please check the section Ethics/specimen and field samples, as I don't think it applies to your study.

4/ Thank you for providing The Paper Explained. During our cross-check analysis, similarities were found between the section "impact" of the TPE and previously published material (see attached). Please modify accordingly.

5/ Thank you for providing a nice synopsis image. Could you please provide a higher resolution version?

6/ As part of the EMBO Publications transparent editorial process initiative (see our Editorial at <http://embomolmed.embopress.org/content/2/9/329>), EMBO Molecular Medicine will publish online a Review Process File (RPF) to accompany accepted manuscripts.

This file will be published in conjunction with your paper and will include the anonymous referee reports, your point-by-point response and all pertinent correspondence relating to the manuscript.

Please let us know whether you agree with the publication of the RPF.

Please also clarify the origin of the figure for reviewer. If it comes from another publication, we suggest having the figure removed at publication stage, with the mention "figure for reviewer only".

I look forward to receiving your revised manuscript.

Yours sincerely,

Lise Roth

**** Reviewer's comments ****

Referee #1:

Is suitable for publication

The authors addressed the remaining editorial issues.

30th Jan 2024

Dear Dr. Dakhel Plaza,

Thank you for submitting the revised files. I am pleased to inform you that your manuscript is accepted for publication and is now being sent to our publisher to be included in the next available issue of EMBO Molecular Medicine!

If you have any questions, please do not hesitate to contact the Editorial Office.

Congratulations on your interesting work!

Yours sincerely,

Lise Roth
